# SSDi8: Accurate and Efficient 8-bit Quantization for State Space Duality

**Hyunwoo Kim**[1]* **Byoungchan Ko**[2]* **Minseok Kang**[2] **Minwoo Kim**[2]
**Dongjin Lee**[3] **Jaehoon Lee**[4] **Sungroh Yoon**[3,4,5]† **Dahuin Jung**[1]†
[1]Department of Artificial Intelligence, Chung-Ang University
[2]School of Computer Science and Engineering, Soongsil University
[3]Department of Electrical and Computer Engineering, Seoul National University
[4]Interdisciplinary Program in Artificial Intelligence, Seoul National University
[5]AIIS, ASRI, INMC, and ISRC, Seoul National University

## Abstract

Recent advances in sequence modeling have highlighted Mamba as a state space architecture offering efficient long-range dependency modeling and providing a viable alternative to Transformers. Building upon this, Mamba-2 introduces the Structured State Space Duality (SSD), which integrates recurrent and attention modes to achieve efficiency and scalability. However, this architectural expansion substantially increases memory and latency overhead, underscoring the need for efficient compression strategies tailored to SSD. In this work, we present SSDi8, the first post-training quantization framework specifically designed for SSD to maintain a persistent INT8 path. SSDi8 introduces a reformulation that decouples element-wise multiplications from matrix multiplications, enabling reuse of quantized activations across modules. Moreover, SSDi8 adaptively quantizes channel-varying activations at cost-effective points, further reducing latency. On the accuracy side, SSDi8 explicitly leverages the intrinsic dimensional decomposition of SSD, exploiting distinct outlier distributions across axes, and incorporates an error correction term based on per-channel error statistics. Comprehensive experiments demonstrate that SSDi8 achieves accuracy comparable to FP16 while delivering up to $1.4\times$ speedup in W4A8 and W8A8 settings. We further validate its robustness in resource-constrained environments by deploying it on the Orin NX device. Code is available at https://github.com/cau-hai-lab/SSDi8.

## 1 Introduction

Mamba (Gu & Dao, 2024) is a recent state space sequence model that builds upon the Structured State Space Model (SSM) (Gu et al., 2020; 2022) to provide efficient long-range dependency modeling with constant computation and memory usage. While global attention in Transformers (Vaswani et al., 2017) can enhance performance as model size increases, it also incurs quadratic growth in computation and memory with respect to sequence length, which poses substantial challenges for large-scale training and deployment. In contrast, Mamba achieves performance comparable to or exceeding state-of-the-art architectures across billion-scale language models, positioning it as a strong candidate for next-generation sequence modeling.

Despite its algorithmic efficiency, Mamba faces practical limitations: its specialized state space recurrence is difficult to parallelize on modern accelerators, making it less hardware-friendly than optimized Transformer kernels, and it shows relatively diminishing efficiency when scaled to larger parameter sizes. To overcome these issues, Mamba-2 (Dao & Gu, 2024) introduces the Structured State Space Duality (SSD), a hybrid design that integrates recurrent mode with attention mode. Mamba-2 adds a head dimension analogous to multi-head attention to enhance scalability and employs a dual representation that improves general matrix multiplication (GEMM) utilization, yielding higher throughput on GPUs and TPUs. While the original Mamba exhibited limited efficiency

---

*Equal Contribution
†Corresponding Authors

beyond 2.7B parameters, Mamba-2 scales effectively to over 8B parameters and achieves competitive performance across language, audio (Lee et al., 2025), vision (Shi et al., 2024), and multimodal tasks (Huang et al., 2024). Yet this expansion also intensifies memory and latency overhead, highlighting the need for efficient compression and optimization.

The recurrent mode of SSD is computationally efficient but system-inefficient, while the attention mode is relatively computationally demanding. During its operation, SSD repeatedly invokes activations across modules and performs sequential updates. In this process, activations reuse across modules necessitates frequent DRAM accesses, and the intrinsically higher latency of DRAM introduces considerable overhead.

As shown in Tab. 1, directly applying quantization methods originally designed for Transformers—such as Hadamard rotation or GPTQ—to SSD layers leads to substantial accuracy degradation. This stems from the distinctive computational organization of SSD. First, the model dimension is partitioned into the number of heads and the per-head dimension, each following markedly different statistical distributions; failure to account for this property results in significant performance loss. Second, SSD contains dimension-varying activations whose shapes differ between memory storage and computation, and these activations are

Table 1: Accuracy under major layer quantization of Mamba-2. Significant degradation arises when SSD is quantized per-tensor.

| Model | Bitwidth | Quantized Layer(s) | ACC |
|---|---|---|---|
| 2.7B | FP16 | – | 63.8% |
| | W4A8 | + In Proj | 63.6% |
| | | + SSD | 58.4% |
| | | + Out Proj | 54.6% |

repeatedly invoked across multiple modules. Third, element-wise multiplications are extensively intertwined with matrix multiplications, further complicating quantization. In this work, we conduct the first comprehensive analysis of SSD to maintain a persistent INT8 path, providing observations that reveal the internal factors contributing to its quantization sensitivity.

Accordingly, we propose SSDi8, an accurate and efficient post-training quantization framework that reduces both inference latency and performance degradation within SSD. For latency reduction, SSDi8 quantizes channel-variant and recurrent activations at optimal points and reuses them, ensuring an uninterrupted INT8 execution path from input to output. Furthermore, we address element-wise operations that disrupt this path by introducing a sparse-aware reformulation, with the guarantee formally established through mathematical analysis. This design keeps the execution in INT8 while substantially alleviating memory bottlenecks and computational overhead. For accuracy, SSDi8 leverages the intrinsic dimensional structure and properties of SSD. Specifically, external dimensions entering SSD are decomposed into two axes, each exhibiting distinct outlier distributions, which are explicitly exploited to reduce quantization error. Furthermore, we introduce an error correction term based on per-channel error means, yielding consistent gains in accuracy. Through these mechanisms, SSDi8 achieves a balanced optimization of both efficiency and performance.

SSDi8 achieves accuracy comparable to FP16 while enabling up to $1.4\times$ inference speedup under both W4A8 and W8A8 configurations, while excluding W4A4 due to hardware-induced slowdowns as discussed in Lin et al. Notably, in the context of SSD—where error sensitivity often causes severe degradation—our method incurs negligible accuracy loss while delivering substantial latency reductions, with single-inference speedups reaching $1.5\times$. To the best of our knowledge, this represents the first successful application of persistent INT8 path within the Mamba-2 SSD architecture. Furthermore, we demonstrate that SSDi8 maintains efficiency in resource-constrained environments through deployment on the Orin NX device.

## 2 RELATED WORK

**Mamba Architecture.** Mamba is a sequence modeling architecture built on SSMs, which has been explored as an alternative to Transformers in order to circumvent the quadratic complexity of self-attention (Gu & Dao, 2024). Unlike conventional linear SSMs (Gu et al., 2022; Smith et al., 2023), Mamba incorporates a selective state space mechanism that adaptively gates input-dependent state transitions and output projections, enabling more expressive sequence modeling. Mamba-2 extends this framework by introducing structured SSDs (Dao & Gu, 2024), which establishes a formal equivalence between SSMs and linear attention and enables optimized GEMM-based implementations. This design substantially improves hardware utilization on modern accelerators. Furthermore, Mamba-2 allows the state dimension—previously constrained to $N = 16$ in Mamba-1—to scale sta-

bly to $N = 64$–$128$ and beyond. In addition, Mamba-2 integrates a multi-head structure analogous to multi-head attention, further enhancing scalability. These advances make large-scale parameter expansion feasible, but they also intensify memory and latency overhead, motivating the need for compression and deployment strategies.

**Quantization for Mamba Models.** Recently, several studies have begun to explore quantization for the Mamba models (Tang et al., 2024; Yu et al., 2025). MambaQuant (Xu et al., 2025) and Quamba1 (Chiang et al., 2025b) introduced Post-Training Quantization (PTQ) methods targeting the original Mamba-1 architecture, but their approaches are not directly applicable to SSD-based Mamba-2. Quamba2 (Chiang et al., 2025a) extended quantization to Mamba-2, applying W4A8 and W8A8 settings that include SSD blocks. However, its method is limited to the inputs of SSD layers and does not adequately address precision issues within internal SSD computations, leaving the INT8 execution path incomplete and constraining latency optimization.

## 3 BACKGROUND

### 3.1 QUANTIZATION

Quantization discretizes continuous values into a finite set of integer levels. In particular, uniform quantization divides the value range into equal intervals, mapping each element of a tensor $X$ to its nearest quantized level as follows:

$$\widetilde{X} = \text{round}\left(\frac{X}{\alpha_X}\right), \qquad \alpha_X = \frac{\max(|X|)}{2^{b-1} - 1}, \tag{1}$$

where $\widetilde{X}$ is the quantized tensor, $\alpha_X$ is the scaling factor that defines the step size based on the maximum absolute value of $X$, and $b$ is the bit-width.

### 3.2 MAMBA-1

Mamba is an architecture built upon State Space Models (SSMs), composed solely of activation operations, where a hidden state variable is employed to efficiently compress and propagate memory (Gu & Dao, 2024). The fundamental state update and output equations are defined as follows:

$$h'(t) = Ah(t) + Bx(t), \quad y(t) = Ch(t). \tag{2}$$

Eq. 2 builds on the theoretical foundations of HiPPO (Gu et al., 2020) and S4 (Gu et al., 2022), which substantially improve both performance and efficiency. However, since SSMs are defined in continuous time, applying them to discrete inputs requires discretization. In practice, Zero-Order Hold is used to preserve previous values, and a time-step activation $\Delta$ is introduced to discretize matrices $A$ and $B$. These operations are performed independently along the channel dimension of the input $x$, so that each channel independently follows its own SSM formulation:

$$S_B(x) = xW_B, \quad S_C(x) = xW_C, \quad S_\Delta(x) = xW_\Delta. \tag{3}$$

Through input-dependent activations, Mamba highlights important information while suppressing noise, improving long-range dependency modeling.

## 4 METHODOLOGY

The overall workflow of SSDi8 is illustrated in Fig. 1. A substantial portion of SSD modules is executed along the persistent INT8 representation path, reusing quantized activations and applying a sparse-aware reformulation to element-wise operations that disrupt this path. The output tensor $dA_{cs}$ from `ChunkCumsum` is negligible in size compared to other tensors, yet its recovery after quantization is challenging due to the element-wise multiplication; hence, it is retained in FP16. In the same vein, `ChunkScan2` remains in FP16 for analogous reasons. These choices are further elaborated within this section.

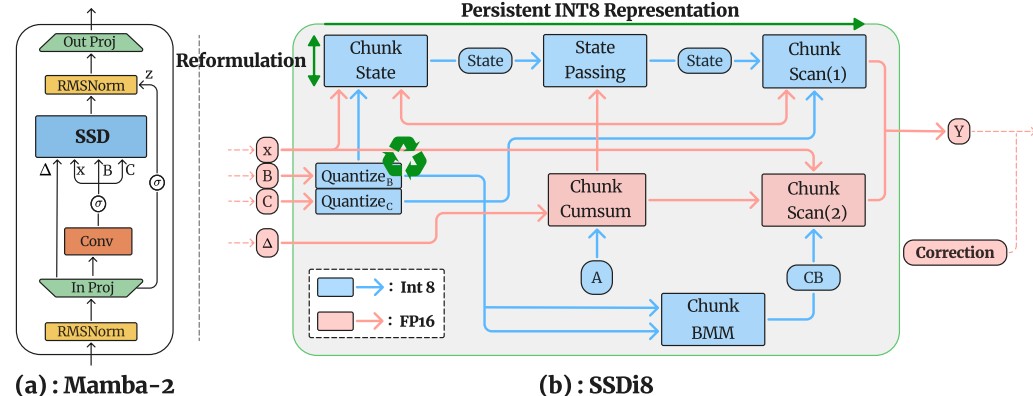

(a) : Mamba-2       (b) : SSDi8

Figure 1: (a) Mamba-2 block architecture. (b) SSD pipeline in SSDi8. SSDi8 enables the persistent INT8 representation path through reformulation and quantized activation reuse, while mitigating performance degradation via channel-aware quantization and mean correction.

## 4.1 PRELIMINARY STUDY: MAMBA-2'S STRUCTURED STATE SPACE DUALITY

The Structured State Space Duality (SSD) in Mamba-2 consists solely of activation operations and unifies the recurrent and attention modes, thereby reducing computational cost and improving efficiency over the recurrence-dominated operations of conventional SSMs. Concretely, the SSM computation can be expressed as a lower-triangular structured matrix: the diagonal block, which directly influences the output, is computed via the attention formulation using matrix multiplications, while the off-diagonal blocks, which require recurrence, are computed by leveraging the semiseparable property, which admits low-rank factorizations.

A key distinction from Mamba is that Mamba-2 introduces a number of heads H, analogous to the multi-head structure in Transformers. As shown in Fig. 2, the value of H is formally defined by $D = H \odot P$, where D denotes the model dimension and P the head dimension. Notably, H and P remain independent axes, with H chosen to be much larger than P. For efficiency, the input-dependent B and C are parameterized with an auxiliary dimension G, and broadcast to H when required.

Formally, the input activations of SSD and its dimension before discretization are given as follows:

$$A \in \mathbb{R}^{(\text{H})}, \qquad \Delta \in \mathbb{R}^{(\text{B,L,H})}, \qquad X \in \mathbb{R}^{(\text{B,L,H,P})},$$

$$B \in \mathbb{R}^{(\text{B,L,G,N})}, \quad C \in \mathbb{R}^{(\text{B,L,G,N})}, \quad Y \in \mathbb{R}^{(\text{B,L,H,P})},$$

where B denotes the batch size, L the sequence length, H the number of heads, G the number of groups, P the head dimension, N the state dimension, and Y the final output of SSD. To shorten the effective recurrent path and enable parallelism, the sequence is partitioned as $L = c \odot l$, where c is the number of chunks and l is the chunk size. The computation then proceeds through five modules—ChunkCumsum, ChunkState, StatePassing, ChunkBMM, and ChunkScan—which together yield the SSD output Y. Additional details are provided in Appendix B .

**ChunkCumsum (Input $(\Delta, A) \mapsto$ Output $(\Delta, dA_{\text{cs}})$).** ChunkCumsum applies a softplus transformation to $\Delta$, a time-step dependent scaling factor introduced for discretization, and discretizes the decay activation $A$ that governs recurrent dynamics. It then prepares the cumulative decay term $dA_{\text{cs}}$, which is subsequently consumed by downstream modules for state updates.

**ChunkState (Input $(dA_{\text{cs}}, \Delta, B, X) \mapsto$ Output $(\text{State})$).** The ChunkState module discretizes the projection matrix $B$, applies the decay factor, and multiplies it with the input $X$ to generate the hidden state. The cumulative decay is computed as $\text{Decay}_{\text{state}} = \exp(dA_{\text{cs}}^{final} - dA_{\text{cs}})$. For simplicity, we denote $\Delta \odot \text{Decay}_{\text{state}}$ by $LUT_{\text{state}}$ where $\odot$ denotes element-wise multiplication, in the following modules. The resulting state update is formulated as

$$\text{State} = X \times (B \odot LUT_{\text{state}}) \tag{4}$$

**StatePassing (Input $(\text{State}, dA_{\text{cs}}) \mapsto$ Output $(\text{State})$).** This module integrates the states computed from independent chunks into the actual recurrent state through decay. The decay term is given

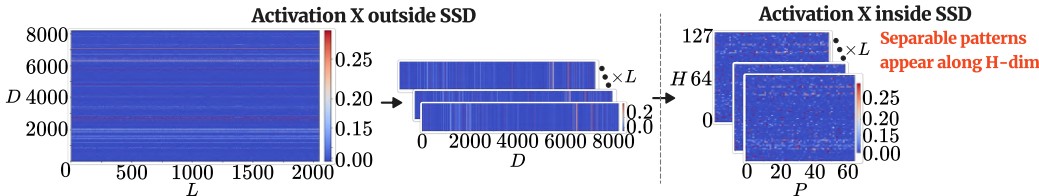

Figure 2: Visualization of activation $X$ in the 16th block of Mamba-2 8B before and after the SSD input transformation. The pre-SSD dimension (`B,L,D`) exhibits no clear token-wise pattern, whereas the transformed dimension (`B,L,H,P`) within SSD reveals distinct patterns along the H-dim.

by

$$\text{Decay}_{\text{pass}} = \exp\big(dA_{\text{cs}}^{final}\big), \tag{5}$$

and the recurrent update is performed over the chunk as

$$\text{State}_c \in \text{State}_{i+1} \leftarrow \text{State}_{i+1} + \text{Decay}_{i+1} \odot \text{State}_i, \qquad i = 0, 1, \dots, c-2. \tag{6}$$

**ChunkBMM (Input** $(B, C) \mapsto$ **Output** $(CB)$**).** `ChunkBMM` performs a batched matrix multiplication between $C$ and $B$. This operation extracts the diagonal blocks of the product, yielding $CB$, which is used in the output computation within SSD.

**ChunkScan1 (Input** $(\text{State}, C, dA_{\text{cs}}, \Delta) \mapsto$ **Output** $(out_{\text{off-diag}})$**).** `ChunkScan1` computes the off-diagonal interaction term by performing a matrix multiplication between the recurrent state State and the projection matrix C. The decay contribution is modeled as $\text{Decay}_{\text{scan1}} = \exp(dA_{\text{cs}})$, and combined with $\Delta$ to form $LUT_{\text{scan1}}$ ($= \Delta \odot \text{Decay}_{\text{scan1}}$). The final off-diagonal output is obtained as $out_{\text{off-diag}} = \big(\text{State} \times C\big) \odot LUT_{\text{scan1}}$.

**ChunkScan2 (Input** $(X, CB, dA_{\text{cs}}, \Delta) \mapsto$ **Output** $(out_{\text{diag}})$**).** `ChunkScan2` computes the diagonal contribution by projecting the input representation $X$ with the combined activation $CB$, while modulating the result using the decay and discretization terms $(dA_{\text{cs}}, \Delta)$. This module complements the off-diagonal pathway from `ChunkScan1`, and together they form the complete output of SSD: $Y = out_{\text{off-diag}} + out_{\text{diag}}$.

### 4.2 SSDI8

**Quantization of B,C.** Within SSDi8, the handling of the channel-dependent activations $B$ and $C$ constitutes one of the strategies, since they are repeatedly invoked across three SSD submodules. Rather than quantizing them separately within each module, SSDi8 quantizes once and reuses the resulting INT8 tensors, thereby reducing memory traffic and enabling a consistent low-precision execution path. A challenge arises because $B$ and $C$ are defined along the group dimension `G` but are broadcast to the head dimension `H` during computation, with `H` typically an order of magnitude larger than `G`. Naively applying quantization after broadcasting induces significant overhead (up to $4\times$), which SSDi8 addresses by optimizing the placement of quantization operations.

To minimize redundant overhead, SSDi8 performs an early quantization of the channel-varying activations $B$ and $C$ once along the group axis `G` at the beginning of each SSD layer. The resulting INT8 tensors are then reused across all downstream modules, maintaining a consistent low-bitwidth representation without repeated quantization. Since $|\text{G}| \ll |\text{H}|$, quantization along `G` is considerably more efficient, adding only about $3\%$ to the total SSD latency. Moreover, as shown in Figs. 2 and 7, the head dimension `H` exhibits highly heterogeneous value distributions across heads—up to $5\times$ variation—making direct per-head quantization unstable, Similarly, the group dimension `G` shows distinct characteristics and must be considered in quantization. While the state dimension `N` exhibits relatively consistent statistics, it directly participates in subsequent matrix multiplications, where quantization errors cannot be restored. Thus, it is excluded from the quantization axes.

**Sparse-aware Reformulation.** As defined in Eq. 4, the `ChunkState` computation applies $B \odot LUT_{\text{state}}$ prior to the matrix multiplication with $X \in \mathbb{R}^{(\text{B,H,c,l,P})}$. Here, $LUT_{\text{state}} \in \mathbb{R}^{(\text{B,H,c,l})}$ is multiplied element-wise with $B \in \mathbb{R}^{(\text{B,H,c,l,N})}$ to impose a decay pattern across the steps within each `B`, `H`, and `c`. The resulting $B \odot LUT_{\text{state}}$ is then multiplied with $X$ along the `l`-axis to project

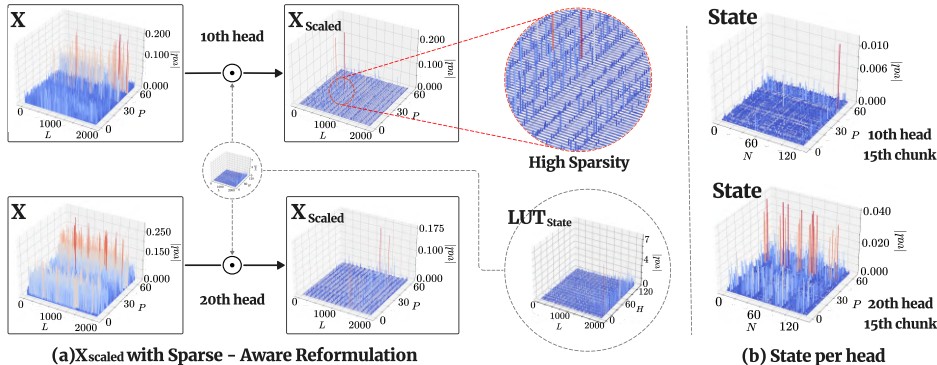

Figure 3: (a) Distribution plots of head-wise $X$ and $LUT_{\text{state}}$ in the 27th block of the `ChunkState` module, and their element-wise product after reformulation $X_{\text{scaled}}$. The channel-wise (`P`-dim) distribution of $X_{\text{scaled}}$ is highly sparse. (b) Head-wise distribution plots of State.

the `l` sequence steps into `N`. The operations are executed independently and in parallel across `B`, `H`, and `c`. However, this ordering introduces three critical limitations: (i) although $B$ is quantized to INT8, the presence of $LUT_{\text{state}}$ in FP16 enforces a floating-point execution path, undermining the efficiency of INT8 GEMM; (ii) because $LUT_{\text{state}}$ exhibits exponential variation along the chunk axis `l`, any quantization scheme other than per-`l` quantization introduces substantial error, while even per-`l` quantization is infeasible due to quantization error accumulation after the $l$-axis matrix multiplication; (iii) attempting $Q(B \odot LUT_{\text{state}})$ requires quantization after the $G \to H$ expansion, which incurs significant overhead. To enable a fully INT8 execution path, SSDi8 reformulates the computation as

$$\text{State}_{\text{INT32}} = Q(X_{\text{scaled}}) \times Q(B), \quad X_{\text{scaled}} = LUT_{\text{state}} \odot X, \qquad \text{State}_{\text{INT32}} \in \mathbb{R}^{(\texttt{B},\texttt{H},\texttt{c},\texttt{P},\texttt{N})}, \quad (7)$$

where $Q(\cdot)$ denotes quantization. This reformulation is valid because $LUT_{\text{state}}$ applies its multiplication along the $l$-dimension shared by both $X$ and $B$, while all other dimensions operate independently. This property ensures that moving the scaling operation from $B$ to $X$ preserves the computational result, and quantizing the resulting $X_{\text{scaled}}$ mitigates the limitations. In this case, $Q(X_{\text{scaled}})$ is quantized along the (`P`, `H`) axes because $LUT_{\text{state}}$ is broadcast along the `P` axis while $X$ preserves consistency across `P` and per-(`H`) heterogeneity as shown in Fig. 3(a) and Fig. 2. Quantization simulations show that $X_{\text{scaled}}$ exhibits pronounced outliers along the channel axis, which makes accurate quantization challenging. However, the actual quantization error of $Q(X_{\text{scaled}})$ does not significantly increase despite the presence of such outliers. From a distributional perspective, this robustness can be attributed to the high sparsity of $X_{\text{scaled}}$ as shown in Fig. 3 (a), which leads to reduced quantization errors overall. To formally validate this property, we prove in Appendix A that, under mild conditions, the quantization error of $X_{\text{scaled}}$ is smaller than that of $Q(X) \odot LUT_{\text{state}}$. This sparsity-aware proof justifies the proposed reformulation, and empirical results further confirm that the resulting performance degradation remains negligible.

**Persistent INT8 Representation of Recurrent States.** $\text{State}_{\text{INT32}}$ obtained from the proposed reformulation is accumulated in INT32. Since INT32 consumes twice the memory of FP16, SSDi8 reduces memory traffic by directly converting INT32 to INT8 in registers with quantization scales:

$$\text{State}_{\text{INT8}} = \text{Round}\left(\text{State}_{\text{INT32}} \odot \frac{s_x s_b q_{\max}}{s_s}\right), \qquad q_{\max} = 2^{b-1} - 1, \qquad (8)$$

where $s_x$, $s_b$, $s_s$ denote the quantization scales of $X$, $B$, and State, respectively. The resulting INT8 tensor is then stored in DRAM, avoiding intermediate FP16 representations and thereby reducing memory bandwidth usage. State also exhibits variation across heads `H`. As shown in Fig. 3 (b), consistency is observed along both the `P` and `N`, since `N` participates in subsequent multiplications within `ChunkScan1`, quantization along `N` is not adopted. $\text{State}_{\text{INT8}}$ is thus quantized per-(`H`,`P`).

In the `StatePassing` module, independently computed chunkwise states are recurrently accumulated with decay to form the actual state, as shown in Eq. 6. Since State is already in INT8, maintaining the INT8 execution path requires quantizing the FP16 Decay. The computation proceeds independently along `B`,`H` and recurrently along `c`, where each Decay is a scalar. This enables

Table 2: Evaluation of Mamba-2 (1.3B, 2.7B, 8B) with three quantization methods (Quamba, Quamba2, and SSDi8) on six zero-shot tasks (LA, HS, PIQA, Arc-E, Arc-C, WG).

| Model | Size | Methods | Bitwidth | LA | HS | PIQA | Arc-E | Arc-C | WG | Avg. |
|---|---|---|---|---|---|---|---|---|---|---|
| Mamba-2 | 1.3B | - | FP16 | 65.6% | 59.9% | 73.3% | 64.1% | 33.3% | 60.8% | 59.5% |
| | | Quamba | W8A8 | 49.8% | 58.5% | 71.2% | 61.9% | 32.1% | 58.1% | 55.2% |
| | | Quamba2 | W8A8 | 62.0% | 59.2% | 72.5% | 63.4% | 32.7% | 60.0% | 58.3% |
| | | | W4A8 | 61.0% | 58.8% | 72.4% | 62.7% | 32.6% | 59.1% | 57.7% |
| | | SSDi8 (Ours) | W8A8 | **64.7%** | **59.7%** | 72.7% | 64.0% | 32.8% | 60.9% | **59.1%** |
| | | | W4A8 | 63.6% | 59.2% | 72.7% | 63.5% | 33.5% | 60.4% | 58.8% |
| | 2.7B | - | FP16 | 69.5% | 66.6% | 76.4% | 69.5% | 36.4% | 64.2% | 63.8% |
| | | Quamba | W8A8 | 52.4% | 60.4% | 71.6% | 62.9% | 33.7% | 58.0% | 56.5% |
| | | Quamba2 | W8A8 | 66.1% | 65.5% | 74.4% | 68.4% | **37.1%** | **63.7%** | 62.5% |
| | | | W4A8 | 65.6% | 65.1% | 74.7% | 68.1% | 36.1% | 62.8% | 62.1% |
| | | SSDi8 (Ours) | W8A8 | **68.3%** | **66.2%** | 75.6% | 69.0% | 36.8% | 63.4% | **63.2%** |
| | | | W4A8 | 67.4% | 65.3% | 75.6% | 68.9% | 35.2% | 63.5% | 62.6% |
| | 8B | - | FP16 | 70.9% | 77.7% | 79.7% | 76.0% | 48.0% | 72.0% | 70.7% |
| | | Quamba | W8A8 | 54.0% | 74.6% | 77.1% | 73.5% | 44.2% | 65.5% | 64.8% |
| | | Quamba2 | W8A8 | 69.8% | **77.8%** | 79.1% | **75.9%** | 46.9% | 69.0% | 69.8% |
| | | | W4A8 | 68.8% | **77.1%** | 79.1% | 75.0% | 46.0% | 68.7% | 69.1% |
| | | SSDi8 (Ours) | W8A8 | **70.4%** | 77.2% | **79.6%** | 75.5% | **47.2%** | **71.2%** | **70.2%** |
| | | | W4A8 | **69.9%** | 76.5% | **79.1%** | 75.4% | 46.2% | 70.6% | 69.6% |

element-wise fixed-point quantization of Decay. Formally,

$$Q(\text{State}_{i+1}) \leftarrow Q(\text{State}_{i+1}) + \frac{Q(\text{Decay}_{i+1})}{S} \odot Q(\text{State}_i), \qquad i = 0, 1, \ldots, c-2, \quad (9)$$

where $S$ is a gating constant chosen as $2^k$ to enable bit-shift operations for minimal latency (with $k = 7$ in experiments). Per-$H, P$ quantization ensures that all $\text{State}_{\text{INT8}}$ across c share a common scale. This allows recurrent updates to be performed by simple bit-shift operations. As a result, $\text{State}_{\text{INT8}}$ can be persisted through `ChunkScan1`, enabling INT8 Tensor Core multiplications with $C_{\text{INT8}}$. Here, $\text{Decay} \in \mathbb{R}^{(\text{B}, \text{H}, \text{c}, 1)}$ aligns with the output $out_{\text{off-diag}} \in \mathbb{R}^{(\text{B}, \text{H}, \text{c}, 1, \text{P})}$, so element-wise multiplication is applied directly after the matrix multiplication.

**Quantization of `ChunkBMM` and `ChunkScan2`.** As shown in Fig. 1, the quantized activations $B_{\text{INT8}}$ and $C_{\text{INT8}}$ are reused in the `ChunkBMM` module. Because both are defined on the group dimension G, the multiplication proceeds without conversion to the head dimension $H$, producing $CB_{\text{INT32}}$. The tensor $CB \in \mathbb{R}^{(\text{B}, \text{G}, \text{c}, 1, 1)}$ is larger than $X$, so its quantization yields substantial memory savings. Similar to `ChunkState`, a single INT32 $\rightarrow$ INT8 step is applied to minimize memory traffic. In `ChunkScan2`, $(LUT_{\text{Scan2}} \odot Q(CB)) \times X$ involves $X$ in FP16, enforcing a floating-point path. Due to its shape, $LUT_{\text{Scan2}}$ is element-wise multiplied with $CB$, making post-quantization recovery difficult and rendering reformulation infeasible due to a shape mismatch with $X$. The dequantization scale of $CB$ is fused into $LUT_{\text{Scan2}}$, reducing overhead while allowing partial FP16 execution. Experiments demonstrate that this process alone yields substantial latency gains.

Leveraging the persistent INT8 representation of recurrent states together with the sparse-aware reformulation and reuse of activation, SSDi8 achieves up to $1.38\times$ speedup overall, with gains reaching $1.6\times$ in the `ChunkScan` module compared to FP16 execution.

**Mean Correction for SSD Quantization Error.** To further mitigate the accumulation of quantization errors across SSD layers, we introduce a per-channel mean correction strategy. Given full-precision and dequantized results $XW = Y \in \mathbb{R}^{N,P}$ and $X'W' = Y' \in \mathbb{R}^{N,P}$, minimizing the error between $Y$ and $Y'$ can be formulated as a least-squares problem, and the optimal correction vector $c^\star$ is given in closed form as the channel-wise mean of the quantization error:

$$E_c = \|Y - (Y' + c)\|_F^2 = \sum_{p=1}^{P} \sum_{i=1}^{N} \left((Y - Y')_{i,p} - c_p\right)^2, \quad c_p^\star = \frac{1}{N} \sum_{i=1}^{N} (Y - Y')_{i,p}. \quad (10)$$

To ensure accurate estimation, we adopt a layer-wise sequential update strategy, enabling subsequent layers to reflect the applied corrections and, thereby, capture activation shifts induced by

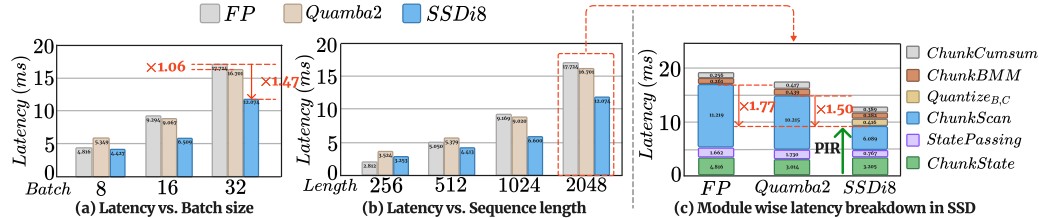

Figure 4: SSD latency of quantization methods on Mamba-2 2.7B: (a) varying batch ($L = 2048$), (b) varying length ($B = 32$), and (c) module-wise latency ($B = 32$, $L = 2048$). PIR denotes Persistent INT8 Representation. SSDi8 achieves up to $1.47\times$ overall speedup and $1.77\times$ in the State path.

earlier updates. For a detailed description of the sequential update algorithm, please refer to Algorithm 1 in Appendix B. To minimize overhead, $c$ is applied only to the output projection layer, whose dimensionality is half that of the input projection layer and where quantization error is most pronounced. This design achieves consistent accuracy gains while incurring only marginal latency overhead ($\approx$ 1–2%).

## 5 EXPERIMENTS

**Experimental Setup.** We conduct PTQ experiments on Mamba-2 (Dao & Gu, 2024) models with 1.3B, 2.7B, and 8B parameters. Experiments are primarily conducted on NVIDIA A5000 GPUs. We evaluate zero-shot performance on LAMBADA (Paperno et al., 2016), WinoGrande (Sakaguchi et al., 2020), PIQA (Bisk et al., 2020), HellaSwag (Zellers et al., 2019), ARC-Easy, and ARC-Challenge (Clark et al., 2018) benchmarks, and additionally assess language modeling capability via WikiText2 perplexity. Results are compared against the FP16 baseline, Quamba (Chiang et al., 2025b) and Quamba2 (Chiang et al., 2025a), and the HAD (HadMamba2) baseline, where HAD applies the Hadamard rotation to the Mamba-2 projection layers (Chiang et al., 2025a), GPTQ weight quantization and RTN quantization of SSD inputs.

**Quantization Setup.** We use symmetric, static quantization on both W8A8 and W4A8 configurations. For 4-bit weight quantization, we employ GPTQ (Frantar et al., 2023), combined with Hadamard-transformed (Ashkboos et al., 2024) projection layers. To handle RMSNorm-induced outliers, we migrate the $\gamma$ parameter (Wei et al., 2022), and apply mean correction with a factor of 0.15 to prevent estimation overfitting.

### 5.1 EVALUATION OF ZERO-SHOT AND GENERALIZATION PERFORMANCE

Tab. 2 reports zero-shot task performance of Mamba-2 models (1.3B, 2.7B, 8B) under FP16, Quamba, Quamba2, and our SSDi8 quantization. Average accuracy is computed over six benchmarks. Across all bit-widths (W8A8, W4A8) and model scales, SSDi8 consistently outperforms Quamba2. For example, on the 2.7B model with W4A8, SSDi8 improves over Quamba2 (62.7% vs. 62.1%), and on the 8B model with W8A8, it achieves 70.2%

Table 3: Wikitext2 perplexity with $L = 2048$.

| Methods | Bitwidth | Wikitext2 Perplexity ($\downarrow$) | | |
| --- | --- | --- | --- | --- |
| | | 1.3B | 2.7B | 8B |
| - | FP16 | 10.42 | 9.06 | 7.25 |
| HAD | W8A8 | 11.31 | 11.42 | 8.57 |
| | W4A8 | 11.63 | 11.85 | 8.79 |
| Quamba2 | W8A8 | 10.80 | 9.32 | 7.79 |
| | W4A8 | 11.08 | 9.54 | 7.94 |
| SSDi8 (Ours) | W8A8 | **10.63** | **9.22** | **7.49** |
| | W4A8 | **10.92** | **9.43** | **7.62** |

compared to 69.8%. These results underscore the robustness of SSDi8 across diverse configurations. Full comparisons, including HadMamba-2 and Quamba2 with W4A16, are provided in Appendix E.

**Perplexity Results.** To assess linguistic fluency and generalization, we report WikiText2 perplexity in Tab. 3. Across all model scales and bit-widths, SSDi8 consistently achieves lower perplexity than Quamba2 while narrowing the gap to FP16. In particular, for the 8B model, SSDi8 yields reductions of 3.9% (7.49 vs. 7.79) under W8A8 and 4.0% (7.62 vs. 7.94) under W4A8. These results demonstrate that SSDi8 preserves linguistic fluency and generalization under quantization.

## 5.2 LATENCY AND MODEL SIZE

In Fig. 4 (a) and (b), we compare SSDi8 with FP16 and Quamba2 on NVIDIA A5000 (24GB) across varying batch sizes ($B \leq 32$) and sequence lengths ($L \leq 2048$). Latency is measured in milliseconds as the average of 100 runs after warm-up. On Mamba-2 2.7B with $B = 32, L = 2048$, SSDi8 achieves a $1.47\times$ speedup over FP16 and a $1.38\times$ improvement over Quamba2. The benefit increases with larger batch sizes and longer sequences, where greater chunk-level parallelism amplifies throughput, while short sequences (e.g., $L = 256$) may show higher FP16 efficiency due to lower computational intensity. Fig. 4 (c) reports module-level latency breakdown for 2.7B at $B = 32, L = 2048$. With persistent INT8 representation, ChunkScan achieves up to $1.77\times$ speedup over FP16 and $1.50\times$ over Quamba2, while StatePassing

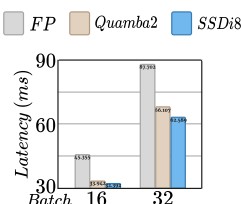

Figure 5: Forward latency of W8A8 ($L = 2048$) on 2.7B.

yields $2.25\times$ and $2.17\times$ improvements, respectively. As demonstrated in Fig. 5, similar gains are observed under W8A8, and results on Mamba-2 8B are provided in Appendix G.

To further assess deployability under resource-constrained conditions, we evaluate SSDi8 on the NVIDIA Orin NX 16G, as shown in Tab. 4. Using the Mamba-2 2.7B model, we measure SSD latency across varying sequence lengths with a batch size of 16, comparing W4A8 and W8A8 quantization against Quamba2. Across all configurations, SSDi8 consistently outperforms Quamba2, demonstrating its robustness beyond high-scale accelerators.

Table 4: SSD latency (ms) of SSDi8 vs. Quamba2 on Orin NX 16G.

| GPU | Orin NX 16G | | | |
|---|---|---|---|---|
| Bitwidth | W4A8 | | W8A8 | |
| Method | Quamba2 | SSDi8 | Quamba2 | SSDi8 |
| $L = 256$ | 55.30 | 44.71 | 51.03 | 41.30 |
| $L = 512$ | 76.10 | 68.00 | 70.95 | 60.49 |
| $L = 1024$ | 134.40 | 127.51 | 139.10 | 114.36 |
| $L = 2048$ | 262.90 | 240.54 | 249.29 | 217.69 |

Additional results evaluating longer sequence lengths and larger batch sizes are reported in Appendix H and Appendix K, respectively.

## 5.3 ABLATION STUDIES

we present ablation results on Mamba-2 2.7B. The baseline retains FP16 only within SSD while applying W4A8 elsewhere. Comparing $Q(X)$ with the proposed reformulated $Q(X \odot LUT_{\text{state}})$ shows negligible quantization error, consistent with our theoretical

Table 5: Ablation results for internal SSD quantization ($Q(\text{SSD})$).

| Bit-width | ChunkState $Q(X)$ | Sparse Reform. | Quant. of B,C | Persistent INT8 | Quant. of ChunkBMM | Latency | PPL |
|---|---|---|---|---|---|---|---|
| | – | – | – | – | – | 8.63 | 9.34 |
| | ✓ | | | | | 8.58 | 9.35 |
| | ✓ | | ✓ | | | 8.05 | 9.37 |
| W4A8 | ✓ | | ✓ | | ✓ | 7.60 | 9.39 |
| | | ✓ | ✓ | ✓ | | 8.35 | 9.36 |
| | | ✓ | ✓ | ✓ | | 8.00 | 9.42 |
| | | ✓ | ✓ | ✓ | ✓ | 6.53 | 9.43 |

proof. Avoiding element-wise multiplications after head expansion of $B$ yields measurable latency gains. Without reformulation, quantizing $X$ alone prevents the use of the persistent INT8 path, and the final latency improvement from quantizing $B$, $C$, and $CB$ is limited to $1.07\times$. By contrast, our reformulation enables INT8 execution in ChunkScan1, improving latency by $1.08\times$, and further quantization of ChunkBMM achieves a $1.32\times$ speedup. Perplexity degradation remains below $0.1$, indicating that our channel-aware quantization preserves accuracy. Further results are provided in Appendix F.

We perform an ablation study on SSD quantization and mean correction using the Lambada dataset, which exhibits minimal performance variance, and report in Tab. 6. On Mamba-2 2.7B under the W4A8 setting, HadMamba quantization yields only 51.2% accuracy, whereas applying SSD quantization substantially boosts performance to 67.2%. Incorporating mean correction provides an additional improvement to 67.4%, achieving consistent accuracy gains with only a ~1–2% overhead. These results demonstrate that SSDi8 achieves both accuracy and efficiency, while mean correction offers effective error correction with negligible additional latency.

Table 6: Ablation results of SSDi8: $Q(\text{SSD})$ and correction $c$.

| Bitwidth | SSDi8 | | Acc. |
|---|---|---|---|
| | $Q(\text{SSD})$ | Correct. | |
| FP16 | – | – | 69.5% |
| W4A8 | | | 51.2% |
| | ✓ | | 67.2% |
| | ✓ | ✓ | 67.4% |

Table 7: Hybrid Mamba–Transformer results on Nemotron-H-8B-Reasoning. SSDi8 is applied only to the SSD path, while other modules remain in FP16.

| Method | Wino | PiQA | ARC-C | ARC-E | Hella | Lamb | Avg. | PPL | SSD Latency (ms) | Fwd. Latency (ms) |
|---|---|---|---|---|---|---|---|---|---|---|
| FP16 | 73.8 | 80.9 | 55.8 | 81.4 | 80.6 | 66.2 | 73.1 | 8.42 | 19.834 | 109.873 |
| INT8 | 73.5 | 80.7 | 55.9 | 81.5 | 80.6 | 66.3 | 73.0 | 8.65 | 9.156 | 98.904 |

## 5.4 RESULTS ON HYBRID MODEL: NEMOTRON-H-8B-REASONING

We report results for applying SSDi8 to the Nemotron-H-8B-Reasoning model in Tab. 7, which adopts a Mamba–Transformer hybrid architecture. In this setting, INT8 quantization is applied exclusively to the SSD path, while all other components are kept in FP16, allowing us to isolate the effect of SSD-path quantization within the hybrid architecture. Tab. 7 includes zero-shot benchmark accuracies and average accuracy, perplexity, as well as SSD-module latency and end-to-end forward latency, enabling a joint assessment of modeling performance and computational efficiency.

Applying INT8 quantization only to the SSD path results in minimal changes in zero-shot performance across Wino, PIQA, ARC-C/E, Hella, and Lamb compared to FP16. The average accuracy decreases slightly from 73.1% (FP16) to 73.0% (INT8), and perplexity exhibits a limited increase from 8.42 to 8.65. In contrast, latency-related metrics show clearer differences. The average SSD-module latency is reduced from 19.834 ms to 9.156 ms, corresponding to approximately a 2× reduction, and the overall forward latency decreases from 109.873 ms to 98.904 ms. These results quantitatively illustrate the contribution of the SSD path to overall inference time and the efficiency gains obtained by INT8 quantization of this component. In this experiment, quantization is restricted to the SSD path, while MLP and attention modules remain in FP16; extending quantization beyond the SSD path is left for future investigation.

## 6 CONCLUSION

In this work, we presented SSDi8, an INT8 quantization framework developed in the context of the SSD of Mamba-2. Unlike prior approaches limited to projections or partial SSD operations, SSDi8 establishes persistent INT8 representations through activation reuse and a sparse-enhanced reformulation. It further explores optimal quantization strategies by analyzing internal activations and incorporates mean correction to compensate for accumulated errors, enabling accurate and efficient inference for large-scale Mamba-2 models. SSDi8 achieves FP16-level accuracy while delivering up to 1.47× speedup over FP16 and 1.38× over Quamba2, and further demonstrates superior efficiency on edge devices such as NVIDIA Orin NX, as well as across diverse batch–sequence settings. SSDi8 provides mathematical intuition for sparse-tensor quantization and offers guidance for quantization in environments where element-wise and recurrent operations are prevalent.

### ACKNOWLEDGMENTS

This work was supported by the Institute of Information & Communications Technology Planning & Evaluation (IITP) grants funded by the Korea government (MSIT) [RS-2021-II211341, Artificial Intelligence Graduate School Program (Chung-Ang University); RS-2021-II211343, Artificial Intelligence Graduate School Program (Seoul National University)], the National Research Foundation of Korea (NRF) grants funded by the Korea government (MSIT) [RS-2025-00555943; 2022R1A3B1077720], the AI Computing Infrastructure Enhancement (GPU Rental Support) User Support Program funded by the Ministry of Science and ICT (MSIT), Republic of Korea (RQT-25-090040), and the BK21 FOUR program (Education and Research Program for Future ICT Pioneers) at Seoul National University in 2025.

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

# APPENDIX

## CONTENTS

# A    PROOF OF PROPOSED QUANTIZATION ERROR REDUCTION VIA REFORMULATION

**Proposition 1.** *Suppose that*

$$\sum_{p=1}^{P} \frac{\Delta_{x,p}^2}{12} \left(\frac{\Delta_{y,p}}{\Delta_{x,p}}\right)^2 \cdot P(y_p \neq 0) \;\; \leq \;\; \|lut\|_2^2 \sum_{p=1}^{P} \frac{\Delta_{x,p}^2}{12}.$$

*Then it holds that*

$$\mathrm{MSE}_{x_{scaled}} \;\; \leq \;\; \mathrm{MSE}_x.$$

**Notation.**
(1) We denote the Hadamard product by $\odot$. The quantization step size is $\Delta = \frac{\text{Range}}{2^b - 1}$.
(2) The dequantized input is $x' = deq(q(x))$. The output is $y_{l,p} = x_{l,p} \odot lut_l$.
(3) Let $\rho_p = P(y_p \neq 0)$ and $y_{l,p}^* = \{y_p : y_p \neq 0\}$.
(4) Vectors are denoted by $x_p = (x_{0,p}, \ldots, x_{L,p})$ and $y_p = (y_{0,p}, \ldots, y_{L,p})$, with error vector $e_{x,p} = (e_{0,x,p}, \ldots, e_{L,x,p})$.
(5) The L-vector $lut = (lut_0, \ldots, lut_L)$ is fixed and deterministic.

**Assumptions.**
(1) $\min(y_p) < 0$ and $\max(y_p) > 0$.
(2) Quantization errors satisfy $e_{x,l,p} \sim U(-\frac{\Delta_{x,p}}{2}, \frac{\Delta_{x,p}}{2})$, $e_{y,l,p} \sim U(-\frac{\Delta_{y,p}}{2}, \frac{\Delta_{y,p}}{2})$.
(3) Outliers are not considered in $y_p$.
(4) $0 < \rho_p < 1$.
(5) $lut$ is not a random variable.

*Proof.* **Step 1. Step size relation.**
In symmetric quantization, the step size $\Delta$ is determined by the min/max values.
By Assumption (1), we have $\Delta_{y,p}^* = \Delta_{y,p}$. Let $s_p = \Delta_{y,p}/\Delta_{x,p}$, so that $\Delta_{y,p} = s_p \Delta_{x,p}$ and hence $\Delta_{y,p}^* = s\Delta_{x,p}$.

**Step 2. Case $y' = (x \odot lut)'$.**
The reconstructed output is

$$y_{l,p}' = \begin{cases} y_{l,p}^* + e_{y,l,p}^*, & \text{with prob. } \rho_p, \\ 0, & \text{with prob. } 1 - \rho_p. \end{cases}$$

Thus

$$\mathrm{MSE}_{x_{\text{scaled}},p} = \rho_p \, \mathbb{E}[(y_{l,p}' - y_{l,p})^2].$$

Since the error $e_{y,l,p}^{\star} = y_{l,p}' - y_{l,p}^{\star}$ has zero mean, we have

$$\mathbb{E}[(e_{y,l,p}^{\star})^2] = \mathrm{Var}(e_{y,l,p}^{\star}).$$

Therefore,

$$\mathrm{MSE}_{x_{\text{scaled}},p} = \rho_p \cdot \mathbb{E}[(e_{y,l,p}^{\star})^2] = \rho_p \cdot \mathrm{Var}(e_{y,l,p}^{\star}).$$

Under the standard quantization noise model,

$$\mathrm{Var}(e_{y,l,p}^{\star}) = \frac{(\Delta_{y,p}^*)^2}{12},$$

so that

$$\mathrm{MSE}_{x_{\text{scaled}},p} = \rho_p \cdot \frac{(\Delta_{y,p}^*)^2}{12}.$$

Averaging over $p$ gives

$$\text{MSE}_{x_{\text{scaled}}} = \frac{1}{P}\sum_{p=1}^{P}\rho_p \frac{(\Delta_{y,p}^*)^2}{12}.$$

**Step 3. Case $y' = x' \odot lut$.**
We expand

$$\text{MSE}_{x,p} = \mathbb{E}\big[\|y_p' - y_p\|_2^2\big] = \mathbb{E}\big[\|(x_p' - x_p) \odot lut\|_2^2\big].$$

By component,

$$\|(x_p' - x_p) \odot lut\|_2^2 = \sum_{l=1}^{L}(e_{x,l,p} \cdot lut_l)^2.$$

Taking expectation,

$$\mathbb{E}[\|e_{x,p} \odot lut\|_2^2] = \sum_{l=1}^{L} lut_l^2 \cdot \mathbb{E}[e_{x,l,p}^2].$$

Since $e_{x,l,p}$ is uniform, $\mathbb{E}[e_{x,l,p}^2] = \Delta_{x,p}^2/12$. Therefore,

$$\text{MSE}_{x,p} = \|lut\|_2^2 \cdot \frac{\Delta_{x,p}^2}{12}.$$

Averaging gives

$$\text{MSE}_x = \frac{1}{P}\sum_{p=1}^{P}\|lut\|_2^2 \frac{\Delta_{x,p}^2}{12}.$$

**Step 4. Comparison.**
Substituting $\Delta_{y,p}^* = s_p\Delta_{x,p}$,

$$\text{MSE}_{x_{\text{scaled}}} = \frac{1}{P}\sum_{p=1}^{P}\rho_p\, s_p^2 \frac{\Delta_{x,p}^2}{12}.$$

Thus, if

$$\sum_{p=1}^{P}\rho_p s_p^2 \frac{\Delta_{x,p}^2}{12} \;\leq\; \sum_{p=1}^{P}\|lut\|_2^2 \frac{\Delta_{x,p}^2}{12},$$

then

$$\text{MSE}_{x_{\text{scaled}}} \leq \text{MSE}_x.$$

$\square$

**Mildness of the sufficient condition.** This condition is *mild*. First, scaling typically reduces the dynamic range so that $\Delta_{y,p} \leq \Delta_{x,p}$, i.e., $s_p \leq 1$. Second, due to the sparsity of $X_{\text{scaled}}$, the activation probability is small ($\rho_p \ll 1$), which diminishes the left-hand side. Third, the $lut$ vector carries non-negligible energy across *dimensions*, so $\|lut\|_2^2$ is not small. Consequently, in these typical regimes,

$$\sum_{p=1}^{P}\rho_p s_p^2 \frac{\Delta_{x,p}^2}{12} \;\leq\; \sum_{p=1}^{P}\|lut\|_2^2 \frac{\Delta_{x,p}^2}{12},$$

and thus $\text{MSE}_{x_{\text{scaled}}} \leq \text{MSE}_x$ follows naturally. For a detailed discussion of the empirical characteristics of the distributions of $x$, $x_{\text{scaled}}$, and $lut$, please refer to Fig. 3 and Appendix L.

# B  ALGORITHM

---

**Algorithm 1** Sequential Mean Correction Update

---

**Require:** Quantized Blocks $B_{1:L}$, fp16 means $\mu_{\text{fp}}[1{:}L]$, number of samples $S$, sequence length $T$ decaying
    factor $\eta$, target-layer set $L_{\text{tgt}}$
    **Fix initial inputs**
 1: **for** $s \leftarrow 1$ **to** $S$ **do**
 2:    $X[s] \leftarrow \text{Embedding}(D[s], T)$
 3: **end for**
 4: **for** $l \leftarrow 1$ **to** $L$ **do**
 5:    **if** $l \notin L_{\text{tgt}}$ **then**
 6:        **for** $s \leftarrow 1$ **to** $S$ **do**
 7:            $Y \leftarrow B_l(X[s])$
 8:            $X[s] \leftarrow Y$
 9:        **end for**
10:        **continue**
11:    **end if**
12:    $\mu_q \leftarrow 0;\ N \leftarrow 0$
13:    **for** $s \leftarrow 1$ **to** $S$ **do**
14:        $Y \leftarrow B_l(X[s])$
15:        $m_s \leftarrow Y.mean(0, 1)$
16:        $n_s \leftarrow Y.shape[0] \cdot Y.shape[1]$
17:        $N \leftarrow N + n_s;\ w_s \leftarrow \dfrac{n_s}{N}$
18:        $\mu_q \leftarrow \mu_q + w_s \cdot (m_s - \mu_q)$
19:    **end for**
20:    $\delta \leftarrow \mu_{\text{fp}}[l] - \mu_q$
21:    $c[l] \leftarrow \eta \cdot \delta$
22:    **for** $s \leftarrow 1$ **to** $S$ **do**
23:        $Y_{\text{comp}} \leftarrow B_l\big(X[s];\ \text{apply } c[l]\big)$
24:        $X[s] \leftarrow Y_{\text{comp}}$
25:    **end for**
26: **end for**
27: **return** model with corrections applied

---

Algorithm 1 implements a layer-wise sequential mean correction strategy. Mean correction estimates a channel-wise correction term from the quantized output mean $\mu_q$, and the accuracy of this estimate depends on the input distribution under which it is measured. If correction terms are estimated independently using a fixed initial input distribution, later layers will observe statistics that differ from those encountered during actual inference, leading to biased correction terms. Instead, we update layers sequentially and propagate corrected activations forward, so that each layer estimates $\mu_q$ under the distribution produced by upstream corrections. This procedure aligns correction estimation with the inference-time signal flow and stabilizes cumulative quantization error across layers.

---

**Algorithm 2** SSD Layer

---

**Require:** $X \in \mathbb{R}(B, L, H, P)$, $\Delta \in \mathbb{R}(B, L, H)$, decay activation $A \in \mathbb{R}(H)$,
 1: $B \in \mathbb{R}(B, L, G, N)$, $C \in \mathbb{R}(B, L, G, N)$,
 2: $L = c \cdot l$

    **Module 1: ChunkCumsum (Input $(\Delta, A) \rightarrow$ Output $(\Delta, dA_{\mathrm{cs}})$)**
 3: $\Delta \leftarrow \mathrm{softplus}(\Delta)$
 4: $A^+ \leftarrow \mathrm{discretize}(A)$
 5: $dA_{\mathrm{cs}} \leftarrow \mathrm{CumSumDecay}(A^+)$                                           $\triangleright \in \mathbb{R}(B, H, c, l)$

    **Module 2: ChunkState (Input $(dA_{\mathrm{cs}}, \Delta, B, X) \rightarrow$ Output State)**
 6: $\mathrm{Decay}_{\mathrm{state}} \leftarrow \exp(dA_{\mathrm{cs}}[:,:,:,l-1] - dA_{\mathrm{cs}})$
 7: $LUT_{\mathrm{state}} \leftarrow \Delta \odot \mathrm{Decay}_{\mathrm{state}}$                                 $\triangleright \in \mathbb{R}(B, H, c, l)$
 8: $\mathrm{State} \leftarrow X \times (B \odot LUT_{\mathrm{state}})$                            $\triangleright \in \mathbb{R}(B, H, c, P, N)$

    **Module 3: StatePassing (Input (State, $dA_{\mathrm{cs}}$) $\rightarrow$ Output State)**
 9: $\mathrm{Decay}_{\mathrm{pass}} \leftarrow \exp(dA_{\mathrm{cs}}[:,:,:,l-1])$                         $\triangleright \in \mathbb{R}(B, H, c)$
10: **for** $i = 0$ to $c-2$ **do**
11:     $\mathrm{State}[i+1] \leftarrow \mathrm{State}[i+1] + \mathrm{Decay}_{\mathrm{pass}}[i+1] \odot \mathrm{State}[i]$
12: **end for**

    **Module 4: ChunkBMM (Input $(B, C) \rightarrow$ Output $CB$)**
13: $CB \leftarrow C \times B$                                         $\triangleright \in \mathbb{R}(B, H, c, l, l)$

    **Module 5: ChunkScan1 (Input $(\mathrm{State}, C, dA_{\mathrm{cs}}, \Delta) \rightarrow \mathbf{out}_{\mathrm{off}}$)**
14: $\mathrm{Decay}_{\mathrm{scan1}} \leftarrow \exp(dA_{\mathrm{cs}})$
15: $LUT_{\mathrm{scan1}} \leftarrow \Delta \odot \mathrm{Decay}_{\mathrm{scan1}}$
16: $\mathrm{out}_{\mathrm{off}} \leftarrow (\mathrm{State} \times C^\top) \odot LUT_{\mathrm{scan1}}$                         $\triangleright \in \mathbb{R}(B, H, c, P, l)$

    **Module 6: ChunkScan2 (Input $(X, CB, dA_{\mathrm{cs}}, \Delta) \rightarrow \mathbf{out}_{\mathrm{diag}}$)**
17: Let $dA_{\mathrm{cs}}^{(m)} \in \mathbb{R}^{(B,H,c,l,1)}$, $dA_{\mathrm{cs}}^{(n)} \in \mathbb{R}^{(B,H,c,1,l)}$ be the broadcasted forms of $dA_{\mathrm{cs}}$.
18: $LUT_{\mathrm{scan2}} \leftarrow \Delta \odot \exp\big(dA_{\mathrm{cs}}^{(m)} - dA_{\mathrm{cs}}^{(n)}\big)$             $\triangleright \in \mathbb{R}(B, H, c, l, l)$
19: $\mathrm{out}_{\mathrm{diag}} \leftarrow X \times (CB \odot LUT_{\mathrm{scan2}})$                  $\triangleright \in \mathbb{R}(B, H, c, P, l)$

    **Final Output**
20: $Y \leftarrow \mathrm{out}_{\mathrm{off}} + \mathrm{out}_{\mathrm{diag}}$
21: **return** $Y$                                          $\triangleright \in \mathbb{R}(B, H, c, P, l)$

---

**SSD layer** computes the state-space dynamics of Mamba-2 using a parallel chunked formulation. Given input activations X, the layer first discretizes the step size $\Delta$ and decay activation A, and constructs per-chunk cumulative decay factors through **ChunkCumsum**.

**ChunkState** performs the input-to-state projection within each chunk in parallel, while

**StatePassing** propagates recurrent information across chunks to restore the global sequence dependency.

**ChunkBMM** computes the block-diagonal interaction matrix CB,

which is exclusively used in the diagonal path.

**ChunkScan1** generates the off-diagonal contribution from the recurrent state, and

**ChunkScan2** produces the diagonal contribution from the input representation with CB.

The final SSD output is obtained by summing these two terms.

## C ADDITIONAL RELATED WORKS

**Post-Training Quantization and LLM Quantization**. Quantization approaches are generally divided into Quantization-Aware Training (QAT) (Gholami et al., 2022), which integrates quantization into the training process, and Post-Training Quantization (PTQ) (Frantar et al., 2023; Xiao et al., 2023; Lin et al., 2024), which applies quantization to models after pretraining. QAT is often considered strong in preserving accuracy, but for large-scale models the associated retraining cost can become prohibitively high. As a result, many recent studies have shifted attention toward PTQ, particularly in the context of large language models (LLMs) (Touvron et al., 2023).

Among representative PTQ approaches, GPTQ (Frantar et al., 2023) proposes a weight-compensation PTQ method by leveraging approximate second-order information via the Hessian. SmoothQuant (Xiao et al., 2023) shifts the difficulty of activation quantization into weights, enabling stable W8A8 and W4A8 performance. QuaRot (Ashkboos et al., 2024) and SpinQuant (Liu et al.) achieve precise 4-bit quantization by applying random or learned rotation matrices to mitigate outliers. QServe (Lin et al.) highlights the practicality of W4A8 quantization in real environments, demonstrating its effectiveness in reducing inference latency for LLMs. However, these methods are inherently optimized for the structural properties of Transformers—such as self-attention and KV caching—and thus are not directly applicable to architectures like selective state space models, where continuous state updates and activation reuse play a central role.

## D ADDITIONAL EXPERIMENTAL SETTING

**Implementation.** For quantization, we use a calibration set of 512 samples drawn from the Pile dataset. We apply 4-bit weight quantization to the in projection and out projection layers using GPTQ. To improve efficiency, the scaling parameter $\gamma$ of RMSNorm is fused into the in projection layer (Wei et al., 2022). Except for the SSD module, activations are quantized to 8-bit with per-tensor quantization, while the fast Hadamard transform (Ashkboos et al., 2024) is fused into the corresponding layers. Inside the SSD, we adopt the same Triton (Dao, 2024b;a) as used in Mamba-2, but modified to fit the SSDi8 method. CUDA (LY, 2024a;b) based causal Conv1d operator is used without modification.

# E  ADDITIONAL ACCURACY RESULTS

Tab. 8 presents an extended version of the accuracy results in Tab. 2. Evaluations are conducted on the same datasets, where HAD denotes applying Hadamard and 4-bit GPTQ quantization to Mamba-2. SSDi8 achieves performance comparable to Quamba2 under W4A16 quantization, even with W4A8 quantization.

Table 8: Evaluation of Mamba-2 models at 1.3B, 2.7B, and 8B scales using four quantization methods—HAD, Quamba, Quamba2, and SSDi8—across six zero-shot tasks: LA, HS, PIQA, Arc-E, Arc-C, and WG.

| Model | Size | Methods | Bitwidth | LA | HS | PIQA | Arc-E | Arc-C | WG | Avg. |
|-------|------|---------|----------|-----|-----|------|-------|-------|-----|------|
| Mamba-2 | 1.3B | - | FP16 | 65.6% | 59.9% | 73.3% | 64.1% | 33.3% | 60.8% | 59.5% |
| | | HAD | W8A8 | 55.3% | 59.4% | 73.2% | 64.0% | 33.5% | 58.2% | 57.3% |
| | | | W4A8 | 53.9% | 58.9% | 72.3% | 63.6% | 33.9% | 59.1% | 57.0% |
| | | Quamba | W8A8 | 49.8% | 58.5% | 71.2% | 61.9% | 32.1% | 58.1% | 55.2% |
| | | Quamba2 | W4A16 | 64.3% | 59.2% | 72.6% | 63.8% | 33.1% | 60.3% | 58.9% |
| | | | W8A8 | 62.0% | 59.2% | 72.5% | 63.4% | 32.7% | 60.0% | 58.3% |
| | | | W4A8 | 61.0% | 58.8% | 72.4% | 62.7% | 32.6% | 59.1% | 57.7% |
| | | SSDi8 (Ours) | W8A8 | 64.7% | 59.7% | 72.7% | 64.0% | 32.8% | 60.9% | 59.1% |
| | | | W4A8 | 63.6% | 59.2% | 72.7% | 63.5% | 33.5% | 60.4% | 58.8% |
| | 2.7B | - | FP16 | 69.5% | 66.6% | 76.4% | 69.5% | 36.4% | 64.2% | 63.8% |
| | | HAD | W8A8 | 53.8% | 60.8% | 73.8% | 64.8% | 35.8% | 62.2% | 58.5% |
| | | | W4A8 | 51.2% | 59.7% | 73.0% | 64.9% | 34.6% | 60.2% | 57.3% |
| | | Quamba | W8A8 | 52.4% | 60.4% | 71.6% | 62.9% | 33.7% | 58.0% | 56.5% |
| | | Quamba2 | W4A16 | 68.8% | 65.6% | 75.5% | 68.6% | 36.6% | 64.9% | 63.3% |
| | | | W8A8 | 66.1% | 65.5% | 74.4% | 68.4% | 37.1% | 63.7% | 62.5% |
| | | | W4A8 | 65.6% | 65.1% | 74.7% | 68.1% | 36.1% | 62.8% | 62.1% |
| | | SSDi8 (Ours) | W8A8 | 68.3% | 66.2% | 75.6% | 69.0% | 36.8% | 63.4% | 63.2% |
| | | | W4A8 | 67.6% | 65.3% | 75.6% | 68.9% | 35.2% | 63.5% | 62.7% |
| | 8B | - | FP16 | 70.9% | 77.7% | 79.7% | 76.0% | 48.0% | 72.0% | 70.7% |
| | | HAD | W8A8 | 56.7% | 75.3% | 78.1% | 74.1% | 45.0% | 65.6% | 65.8% |
| | | | W4A8 | 56.1% | 74.6% | 77.3% | 73.8% | 44.5% | 66.0% | 65.4% |
| | | Quamba | W8A8 | 54.0% | 74.6% | 77.1% | 73.5% | 44.2% | 65.5% | 64.8% |
| | | Quamba2 | W4A16 | 71.2% | 76.8% | 79.1% | 75.2% | 45.9% | 70.8% | 69.8% |
| | | | W8A8 | 69.8% | 77.8% | 79.1% | 75.9% | 46.9% | 69.0% | 69.8% |
| | | | W4A8 | 68.8% | 77.1% | 79.1% | 75.0% | 46.0% | 68.7% | 69.1% |
| | | SSDi8 (Ours) | W8A8 | 70.4% | 77.2% | 79.6% | 75.5% | 47.2% | 71.2% | 70.2% |
| | | | W4A8 | 69.9% | 76.5% | 79.1% | 75.4% | 46.2% | 70.6% | 69.6% |

We evaluate perplexity on the Pile benchmark for the 1.3B and 2.7B models. Across both model scales, SSDi8 surpasses Quamba2 and approaches FP16-level performance under W8A8 quantization, as shown in Tab. 9.

Table 9: Pile perplexity with $L = 2048$

| Model | Methods | Bitwidth | Pile Perplexity ($\downarrow$) | |
|---|---|---|---|---|
| | | | 1.3B | 2.7B |
| Mamba-2 | - | FP16 | 6.99 | 6.27 |
| | HAD | W8A8 | 7.46 | 7.77 |
| | | W4A8 | 7.87 | 8.17 |
| | Quamba2 | W8A8 | 7.20 | 6.44 |
| | | W4A8 | 7.55 | 6.68 |
| | SSDi8 (Ours) | W8A8 | 7.08 | 6.34 |
| | | W4A8 | 7.41 | 6.57 |

# F ADDITIONAL ABLATION STUDIES

Tab. 10 shows ablation results on the quantization axis of activations within SSD, evaluated on Wikitext2 perplexity. For activations $B, C$, per-$\mathtt{G}$, $\mathtt{N}$ yields the best performance, though the difference from per-$\mathtt{G}$ is negligible (0.02). In contrast, $X$ and State are highly sensitive to the choice of quantization axis, showing substantial degradation when either the $\mathtt{P}$ or $\mathtt{H}$ axis is not considered.

Table 10: Ablation study for quantization axis.

| Model | Bitwidth | Activation | per-T | per-P($\mathtt{N}$) | per-H($\mathtt{G}$) | Wikitext2 Perplexity |
|---|---|---|---|---|---|---|
| | | – | – | – | – | 7.42 |
| | | | v | | | 7.59 |
| | | B,C | | v | | 7.43 |
| | | | | | v | 7.44 |
| 8B | W4A8 | | | v | v | 7.42 |
| | SSD-FP16 | | v | | | 11.97 |
| | | X,State | | v | | 8.59 |
| | | | | | v | 8.15 |
| | | | | v | v | 7.42 |

We also analyze latency and accuracy variations with respect to the placement of mean correction. The highest accuracy gain is observed when mean correction is applied immediately after SSD layers, indicating error accumulation within SSD. In Mamba-2, the model dimension is halved after the out-projection layer, yielding the lowest latency when mean correction is applied. Considering the trade-off between latency and accuracy, we therefore apply mean correction only at the out-projection layer, as summarized in Tab. 11.

Table 11: Accuracy and speedup for W4A8. Experiments are conducted on the LAMBADA dataset, using SSDi8 without mean correction as the baseline.

| Bitwidth | Project | Speedup | Acc. |
|---|---|---|---|
| | None | $\times 1.00$ | 67.2% |
| W4A8 | In | $\times 0.945$ | 67.4% |
| | SSD | $\times 0.975$ | 67.5% |
| | Out | $\times 0.987$ | 67.4% |

## G    ADDITIONAL LATENCY AND MODEL SIZE RESULTS

We further analyze the memory footprint and SSD latency of SSDi8, using FP16 and Quamba2 as reference baselines.

Table 12: Memory usage comparison

| Model | Size | Method | W8A8 | W4A8 |
|---|---|---|---|---|
| Mamba2 | 2.7B | FP16 | 5.154 GB | |
| | | Quamba2 | 2.948GB | 1.766GB |
| | | SSDi8 (Ours) | 2.953GB | 1.774GB |
| | 8B | FP16 | 15.710 GB | |
| | | Quamba2 | 9.860GB | 7.028GB |
| | | SSDi8 (Ours) | 9.867GB | 7.038GB |

As shown in Table 12, the memory usage of SSDi8 is nearly identical to Quamba2. For the 2.7B model under W8A8, SSDi8 requires 2.953GB compared to 2.948GB for Quamba2, resulting in only a +0.17% increase due to additional static quantization scales. A similar trend is observed for the 8B model (9.867GB vs. 9.860GB, +0.07%).

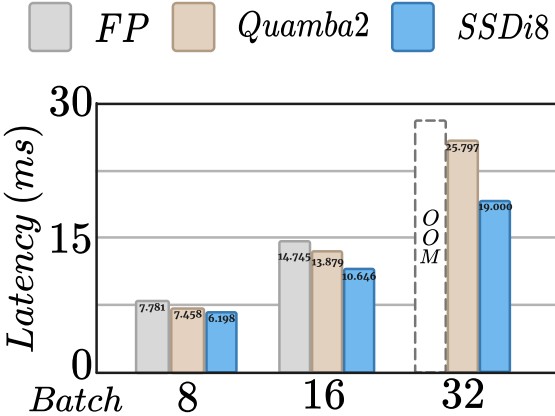

Figure 6: Comparison with Quamba2 under W4A8 quantization on the 8B model is also reported. OOM denotes Out-Of-Memory.

Figure 6 reports SSD-module latency across batch sizes. Compared to FP16, SSDi8 consistently reduces latency at moderate batch sizes. At batch 16, SSDi8 reduces latency from 14.745ms (FP16) to 10.646ms, corresponding to a 27.8% reduction. At batch 32, FP16 encounters OOM, while Quamba2 requires 25.797ms. Under the same setting, SSDi8 achieves 19.000ms, yielding a 26.3% latency reduction compared to Quamba2.

# H    LONGER CONTEXT RESULTS

We report additional results on long-context evaluation beyond 2k tokens(Tab. 13), comparing FP16, Quamba2, and SSDi8 under identical experimental settings. All experiments are conducted on an A5000 GPU using the 2.7B Mamba-2 model with a batch size of 8 and sequence lengths ranging from 2k to 14k. For each context length, we measure SSD-module latency, end-to-end throughput (tokens/s), and perplexity (PPL) to assess both computational efficiency and modeling behavior in the long-sequence regime.

Table 13: Long-context results (B=8): SSD latency, end-to-end throughput (tokens/s), and perplexity (PPL).

| Length | Batch | FP16 | | | Quamba2 | | | SSDi8 | | |
|--------|-------|------|--|--|---------|--|--|-------|--|--|
| | | SSD Lat. (ms) | Throughput (T/s) | PPL | SSD Lat. (ms) | Throughput (T/s) | PPL | SSD Lat. (ms) | Throughput (T/s) | PPL |
| 2k | B=8 | 4.990 | 10480 | 9.062 | 5.313 | 11850 | 9.549 | 4.507 | 11981 | 9.470 |
| 4k | B=8 | 9.299 | 10716 | 13.845 | 8.998 | 12333 | 11.502 | 7.231 | 12818 | 9.021 |
| 6k | B=8 | 13.564 | 10922 | 113.407 | 13.363 | 12648 | 100.438 | 10.367 | 13038 | 8.943 |
| 8k | B=8 | 17.867 | 11173 | 444.467 | 15.932 | 13079 | 309.182 | 12.635 | 13418 | 9.040 |
| 10k | B=8 | 22.061 | 11167 | 1577.003 | 19.806 | 13121 | 1253.093 | 15.827 | 13804 | 9.122 |
| 12k | B=8 | 27.058 | 10887 | 2650.420 | 23.531 | 13079 | — | 18.168 | 13743 | 9.239 |
| 14k | B=8 | 30.870 | 11362 | 3881.000 | 27.368 | 13291 | — | 21.138 | 14200 | 9.360 |

Across all evaluated sequence lengths, SSDi8 consistently achieves higher throughput and lower SSD latency than both FP16 and Quamba2, with the performance gap widening as the context length increases. At 14k tokens, SSDi8 improves throughput from 11,362 tokens/s (FP16) to 14,200 tokens/s, corresponding to a gain of +2,838 tokens/s (+25.0%). Over the same setting, SSD latency is reduced from 30.870 ms to 21.138 ms, yielding a 9.73 ms reduction (-31.5%). In terms of perplexity, FP16 Mamba-2 exhibits rapid degradation beyond a context length of 4k, with PPL increasing from 13.845 (4k) to 113.407 (6k) and further escalating at longer contexts. Quamba2 shows a similar instability trend. In contrast, SSDi8 maintains stable perplexity across the entire evaluated range (2k–14k), with PPL remaining within a narrow band of 8.94–9.36. These results indicate that SSDi8 preserves numerical stability in the long-context regime while simultaneously improving computational efficiency. The stability of SSDi8 at extended sequence lengths suggests that careful quantization of the SSD path mitigates the long-sequence numerical issues observed in full-precision and prior quantized baselines.

# I  CALIBRATION SENSITIVITY ANALYSIS

We analyze the sensitivity of SSDi8 to calibration settings. Specifically, we consider both the number of calibration samples and the choice of calibration dataset. First, we report zero-shot performance while varying the calibration sample size from 128 to 2048 under an identical evaluation protocol, in order to examine how the performance changes as the amount of calibration data varies(see Tab. 14). We then compare performance across different calibration datasets, including commonly used corpora such as C4, PTD, and Pile, to study the effect of the calibration data source (Tab. 15).

Table 14: Sensitivity to the number of calibration samples. Accuracy (%) on six zero-shot tasks.

| Calib Size | Wino | PiQA | ARC-C | ARC-E | Hella | Lamb | Avg. |
|---|---|---|---|---|---|---|---|
| 128 | 64.4 | 75.5 | 35.1 | 68.2 | 65.0 | 68.0 | 62.7 |
| 256 | 64.5 | 75.3 | 36.4 | 67.8 | 64.8 | 67.0 | 62.6 |
| 512 | 64.2 | 75.5 | 35.8 | 67.9 | 65.1 | 67.7 | 62.7 |
| 1024 | 64.3 | 75.7 | 36.4 | 67.6 | 64.8 | 66.3 | 62.5 |
| 2048 | 64.8 | 75.1 | 36.2 | 68.1 | 65.2 | 67.1 | 62.7 |

SSDi8 shows only small variations with respect to both the number of calibration samples and the choice of calibration dataset. When increasing the calibration sample size from 128 to 2048, the average accuracy across six zero-shot tasks remains largely stable, with only minor differences across individual benchmarks. Similarly, changing the calibration dataset leads to comparable performance, and no single calibration corpus consistently results in lower accuracy. Overall, these results suggest that the performance of SSDi8 remains stable across a range of calibration configurations.

Table 15: Sensitivity to the choice of calibration dataset. Accuracy (%) on six zero-shot tasks.

| Calib Data | Wino | PiQA | ARC-C | ARC-E | Hella | Lamb | Avg. |
|---|---|---|---|---|---|---|---|
| C4 | 65.4 | 75.2 | 36.1 | 68.5 | 65.3 | 67.7 | 63.0 |
| PTD | 65.1 | 75.1 | 36.5 | 68.6 | 65.5 | 67.6 | 63.0 |
| Pile | 64.5 | 75.3 | 36.4 | 67.8 | 64.8 | 67.0 | 62.6 |

## J    Latency breakdown of the entire Mamba-2 block

Tab. 16 presents a module-wise latency breakdown of a single Mamba-2 block, decomposing the block into its major computational components: input projection (in-proj), convolution (conv), SSD, normalization (norm), output projection (out-proj), and the correction term. The results are reported for the 55th block of the 2.7B Mamba-2 model under batch sizes 8, 16, and 32, and compare FP16 and SSDi8 under identical experimental settings. This breakdown is intended to quantify where latency reductions are achieved by SSDi8 and to clarify the contribution of each submodule to the overall block latency.

Table 16: Latency breakdown (ms) of the Mamba-2 block (2.7B, $L = 2048$) under different batch sizes.

| Method | Batch | In-Proj | Conv | SSD | Norm | Out-Proj | Correction |
|--------|-------|---------|------|-----|------|----------|------------|
| FP16 | B=8 | 9.829 | 0.781 | 4.795 | 1.196 | 4.919 | – |
| FP16 | B=16 | 20.759 | 1.357 | 9.426 | 1.640 | 10.272 | – |
| FP16 | B=32 | 41.297 | 2.526 | 17.640 | 3.086 | 20.474 | – |
| SSDi8 | B=8 | 8.124 | 0.587 | 4.481 | 1.089 | 3.266 | 0.426 |
| SSDi8 | B=16 | 16.758 | 1.011 | 7.293 | 1.442 | 6.677 | 0.698 |
| SSDi8 | B=32 | 33.893 | 1.799 | 13.199 | 2.291 | 12.914 | 1.252 |

under FP16 execution, the projection layers constitute the dominant latency bottleneck, followed by the SSD computation, while normalization and other components contribute relatively little. In SSDi8, Hadamard-based activation quantization and GPTQ weight quantization are applied to the projection layers, while the primary optimization focus is placed on the SSD module, which represents the second-largest bottleneck in FP16. Across batch sizes, SSD latency increases approximately linearly, exhibiting a scaling trend similar to that of the projection layers, whereas the remaining submodules remain negligible. We further observe that the output projection reduces the dimensionality of the hidden representation, allowing the mean-correction term introduced by SSDi8 to be applied at minimal cost. As a result, the correction latency remains lower than that of the convolution layer, which is the least expensive FP16 component. Overall, this breakdown illustrates how SSDi8 reduces latency primarily by optimizing the most time-consuming components within a Mamba-2 block, while preserving favorable scaling behavior across batch sizes.

# K  BATCH-SIZE SENSITIVITY ANALYSIS

To further examine the computational behavior of SSDi8, we evaluate SSD-module latency and throughput across varying batch sizes, ranging from edge-oriented small batches to large-batch cloud-serving scenarios. This analysis allows us to assess how INT8 quantization of the SSD path scales under different levels of computational intensity (see Tab. 17).

Table 17: SSD latency and throughput comparison ($L = 2048$)

| Batch Size | Method | Value | |
| --- | --- | --- | --- |
| | | Latency (ms) | Throughput (T/s) |
| 16 | FP16 | 2.757 | 9918 |
| | SSDi8 | 2.671 | 10299 |
| 32 | FP16 | 4.898 | 10653 |
| | SSDi8 | 4.510 | 11976 |
| 64 | FP16 | 9.271 | 10405 |
| | SSDi8 | 7.173 | 12114 |
| 128 | FP16 | 18.003 | 11222 |
| | SSDi8 | 13.406 | 13407 |
| 256 | FP16 | 35.194 | 11333 |
| | SSDi8 | 23.730 | 13798 |

SSDi8 consistently reduces SSD-module latency and improves throughput across all batch sizes. At batch size 16, SSD latency decreases from 2.757 ms to 2.671 ms (3.1% reduction), while throughput improves by 3.8%. As batch size increases, the performance gap widens. At batch size 128, SSD latency is reduced by 25.5% and throughput increases by 19.5%. In the large-batch regime (256), SSDi8 achieves a 32.6% reduction in SSD latency (35.194 ms $\rightarrow$ 23.730 ms) and a 21.8% improvement in throughput.

These results indicate that the benefits of quantizing ChunkState and ChunkBMM become increasingly pronounced under higher computational intensity. While small-batch settings already show consistent gains, larger batch sizes amplify the arithmetic and memory-efficiency advantages of INT8 execution. Overall, SSDi8 demonstrates favorable scaling behavior across both edge and cloud-serving deployment regimes.

## L    DISTRIBUTIONS OF SSD TENSORS

**Visualization of Activations.** Figure 7 represents that visualization of $B$, $C$, and $CB$ by group in the first, middle, and last blocks of the Mamba-2 8B model. As argued in Sec. 4, the distributions differ across groups. $CB$ is masked as it is used for computing $out_{\text{diag}}$.

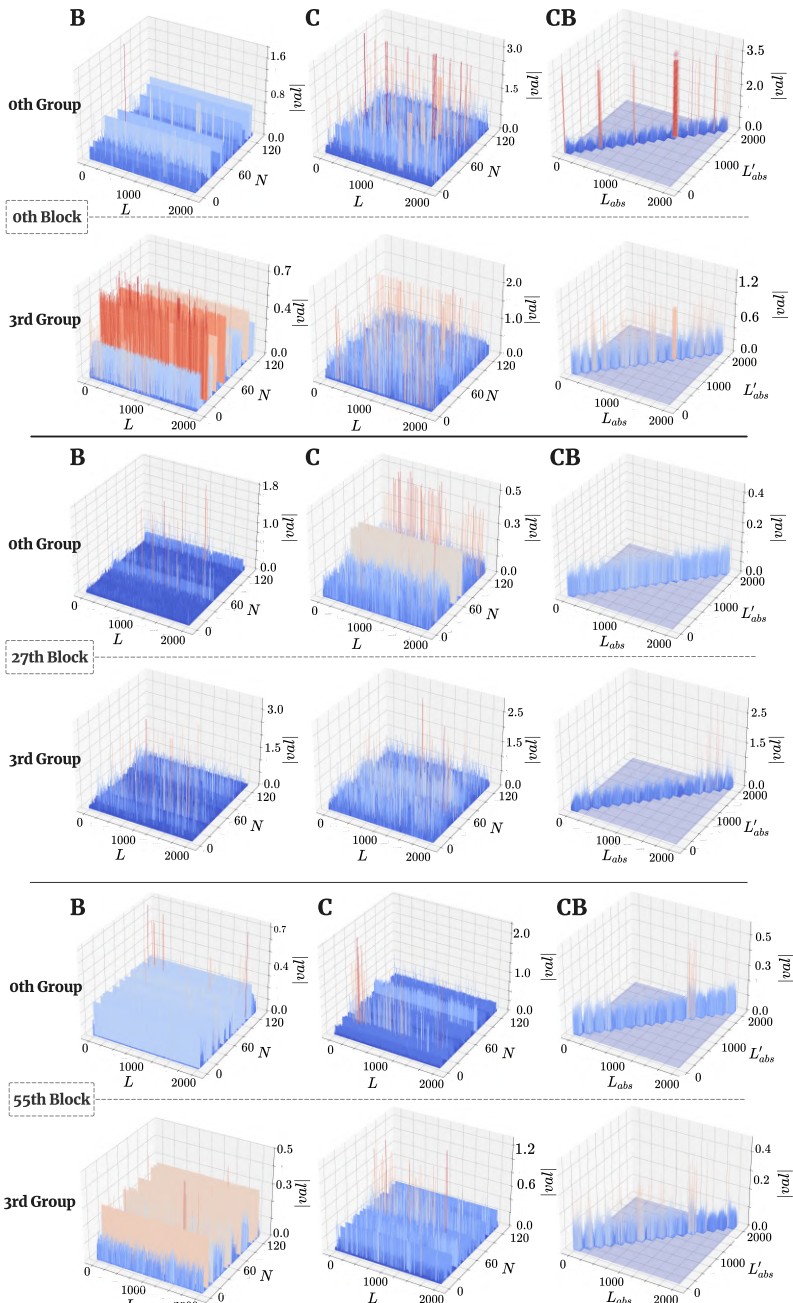

Figure 7: Visualization of the distributions of activations $B$, $C$, and $CB$ in in the first, middle, and last block of Mamba-2 8B.

Figure 8 shows the visualization of $X$, $LUT_{state}$, and $X_{scaled}$ in the last block of Mamba-2 8B. The first row illustrates the full sequence length, while the second row depicts its partition into nchunks with the corresponding chunk size. Both $LUT_{state}$ and $X_{scaled}$ exhibit exponential growth as the chunksize index increases.

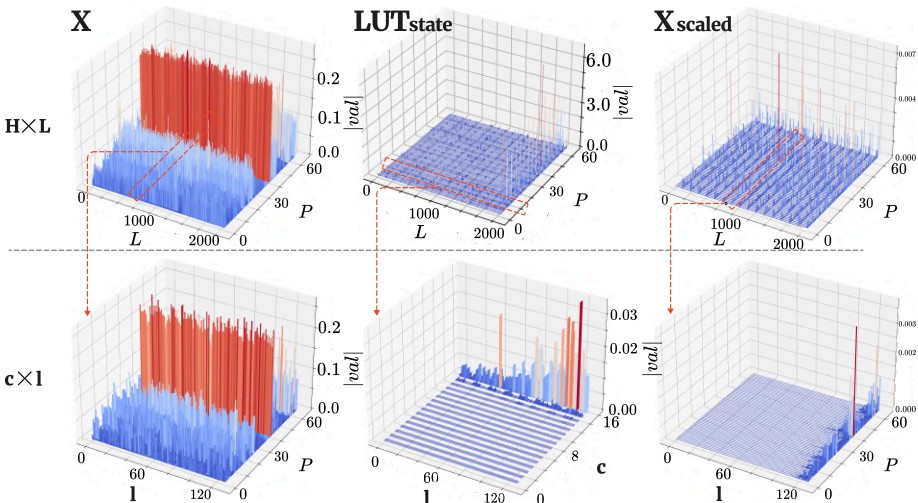

Figure 8: Visualization of the distributions of activations $X$, $LUT_{state}$, and $X_{scaled}$ in the last block of Mamba-2 8B.

## M LLM USAGE

During the manuscript preparation, we used OpenAI's GPT5 (https://chatgpt.com/), a Large Language Model, to proofread our work. Our interaction with the LLM was iterative and focused exclusively on improving the quality of the writing. We affirm that the LLM served as an assistive tool and did not contribute to core research ideas, experimental design, analysis, and results presented in this paper. The final scientific content and all claims made in this paper are the sole responsibility of the authors.

