# OpenReview forum: "SSDi8: Accurate and Efficient 8-bit Quantization for State Space Duality"
_ICLR.cc/2026/Conference — ICLR 2026 Poster_

### Official Review · Reviewer_9vyW · 2025-10-19

**Soundness:** 3
**Presentation:** 2
**Contribution:** 3
**Rating:** 4
**Confidence:** 5

**Summary:**

The paper introduces SSDi8, a framework designed to accelerate Structured State Space Duality (SSD) computation using INT8 representation. Unlike prior methods (e.g., Quamba2) that quantize only the SSD inputs while performing state updates in FP16, SSDi8 establishes a persistent INT8 execution path by quantizing internal SSD computations, reusing quantized activations, and reformulating element-wise operations to ensure low-precision consistency. The framework further incorporates a sparse-aware reformulation to mitigate quantization errors and a mean-correction mechanism to compensate for layer-wise activation drift. Experiments on Mamba-2 models (1.3B, 2.7B, and 8B) demonstrate that SSDi8 achieves up to 1.47× speedup over FP16 while maintaining accuracy comparable to full precision across six zero-shot benchmarks. Moreover, SSDi8 exhibits robust performance on edge devices (e.g., NVIDIA Orin Nano), highlighting its practical deployability under resource constraints.

**Strengths:**

SSDi8 provides a approach to quantizing Mamba-2’s SSD modules with 8-bit, addressing quantization sensitivity and memory inefficiency not handled by prior work. Its sparse-aware reformulation and activation reuse are both theoretically grounded and empirically validated, enabling a full INT8 inference pipeline without significant accuracy loss. The paper includes comprehensive experiments across multiple model scales, showing consistent improvements over baselines like Quamba2 in both accuracy and latency. Furthermore, the mean-correction mechanism is elegantly simple yet effective, improving quantized results with minimal overhead. Finally, the inclusion of both theoretical proofs and deployment evaluations (e.g., Orin Nano) enhances the paper’s completeness and real-world relevance.

**Weaknesses:**

### Major Concerns
- The paper is somewhat difficult to follow, particularly in the presentation of the Sparse-aware Reformulation, which appears to be one of the main contributions. I recommend the authors derive Equation (7) explicitly from the original ChunkState formulation to establish a clearer connection between the reformulated SSD equations and the preceding context.
- The source of the reported accuracy and latency improvements is unclear. To the best of my understanding, Quamba2 [1] already quantizes inputs (x), B, and C to INT8, then dequantizes within the SSM module to perform state computation in FP16. The authors should clarify how SSDi8 maintains state propagation and accumulation in INT8 precision, and why this results in both higher accuracy and lower latency compared to Quamba2. A detailed explanation or ablation isolating the impact of full INT8 state computation would greatly strengthen the technical claims.
- Some related methods, such as HadMamba, are mentioned without citation or discussion. Properly citing these works and clarifying their relationship to SSDi8 would improve the completeness of the related-work section.





[1] Quamba2: A Robust and Scalable Post-training Quantization Framework for Selective State Space Models

**Questions:**

- Could the authors provide a deeper comparison with Quamba2 [1]? In particular, it would be helpful to include Quamba2 results in Table 5 for a more direct and quantitative comparison. If my understanding is correct, Quamba2 already supports ChunkState quantization Q(X) and quantization of matrices B and C. A side-by-side evaluation would better clarify the incremental benefits of the proposed framework over this closely related baseline.
- Would the authors also provide a more detailed latency breakdown of the entire SSD block? Specifically, analyzing the latency contributions of the input projection, SSD computation, normalization, and output projection layers under different input batch sizes would offer valuable insight into the scalability and bottlenecks of the proposed quantization method.

I am willing to reconsider and potentially raise my overall score if the above concerns are adequately addressed in the authors’ rebuttal or follow-up discussion.


### Minor Suggestions
- I recommend that the authors simplify the shape conversion and tensor dimension descriptions in the SSD computation steps. The current presentation is dense and difficult to follow, and a clearer explanation (potentially with a schematic or pseudo-code) would improve readability.
- I suggest that the authors include a discussion on large-batch serving scenarios, as quantization behavior and latency characteristics can differ significantly between small-batch (edge inference) and large-batch (cloud serving) settings. Highlighting how SSDi8 performs under these conditions would enhance the paper’s practical relevance and deployment perspective.

---

> ### Author Response · Authors · 2025-11-21
> **Response #1**
>
> We appreciate the reviewer’s positive evaluation of our work. As noted, SSDi8 offers an effective 8-bit quantization approach for Mamba-2’s SSD modules, addressing sensitivity and memory inefficiencies of SSD that prior methods do not resolve. We are grateful for the reviewer’s recognition of the theoretical and empirical grounding of our Sparse-aware Reformulation and activation reuse, which together enable a full INT8 inference pipeline with minimal accuracy loss. We also appreciate the reviewer’s comments on the completeness of our study—including comprehensive multi-scale experiments, the simplicity and effectiveness of the mean-correction mechanism, and real-world deployment results such as those on Orin Nano.
>
> We address all of your comments in the following responses:
>
> **[W1] Difficult to follow the presentation of the Sparse-aware Reformulation**
>
> Thank you for the valuable suggestion that helped us improve the clarity of our work. Based on your recommendation, we have added a more detailed explanation of the matrix multiplication process and the justification for why the reformulation is valid. The corresponding updates are highlighted in blue in Sec. 4.2.
>
> **[W2, Q1] Why both higher accuracy and lower latency compared to Quamba2?**
>
> | Model (2.7B, B=32, L=2048) | Q(X) | SAR | Q(B,C) | Q(BMM) | PIR | SSD Latency (ms) | Throughput (T/s) | PPL |
> |-----------------------------|------|-----|--------|--------|-----|-------------------|------------------|------|
> | SSD-FP16                   | —    | —   | —      | —      | —   | 14.385            | 12943            | 9.40 |
> | Quamba2                | v    |     | v      |        | X   | 16.014            | 13060            | 9.54 |
> |                            | v    |     | v      | v      | X   | 13.341            | 13609            | 9.55 |
> |                             |      |     |        |        |     |                   |                  |      |
> | Q(SSD)                 |     | v    |        |        |     | 14.409            | 13262            | 9.42 |
> |                            |     | v   |   v     |        |     | 15.724            | 12851            | 9.45 |
> |                            |     | v   |   v     |       |  v   | 12.886            | 13649            | 9.48 |
> |                            |    | v   | v      | v      |  v   | 11.596            | 13823            | 9.48 |
>
> We provide a direct ablation comparing Quamba2 with the SSD—only quantization variants of SSDi8 (Q(SSD)), focusing on ChunkState and ChunkBMM—the key components where the two approaches differ. In the ablation, we denote Sparse-aware Reformulation as SAR and persistent INT8 recurrence as PIR.
> Quamba2 quantizes $X$, $B$, and $C$ at the convolution output and feeds them into SSD in INT8 form. As a result, both SAR and PIR are infeasible: (1) SAR requires quantizing $(X \cdot LUT)$ before expansion, which Quamba2 cannot access; (2) PIR requires keeping the ChunkState output in INT8 until the end of the SSD block, but Quamba2 dequantizes before internal SSD operations. Thus, in Quamba2, we can only evaluate the configurations that quantize $X$ and $(B,C)$ individually.
>
> When examining the original Quamba2 method alone (the first row corresponding to Quamba2), the overall model throughput increases compared to SSD-FP16; however, the SSD-layer latency itself increases due to the dequantization cost and the lack of quantization applied to the internal modules. Similarly, in the second row of Q(SSD)—where quantization is applied to $(X \cdot LUT)$ and to $B$ and $C$, but these quantized values are not actually utilized—the SSD latency also increases relative to SSD-FP16, consistent with Quamba2. In terms of perplexity, the per-group quantization used by Quamba2 results in a degradation of $0.14$, whereas the per-$P,H$ quantization applied in Q(SSD) yields a smaller perplexity degradation of 0.05, thereby minimizing quantization error. When additionally applying one of SSDi8’s methods to Quamba2—specifically, quantizing Q(ChunkBMM) using Q($B$) and Q($C$) (the second row corresponding to Quamba2)—we observe an increase of approximately 600 t/s in throughput and an improvement of about 2.7 ms in latency. The perplexity increase in this case is only around 0.01, which is negligible. However, because SAR and PIR cannot be utilized, no further latency reduction is achievable.
>
> In contrast, when SAR and PIR are enabled and the reuse strategy is applied to obtain Q(ChunkBMM) without additional overhead (the last row corresponding to Q(SSD)), latency improves by approximately 4.2 ms and throughput increases by about 1000 t/s relative to the second-row Q(SSD) configuration. Moreover, the final accuracy degradation is limited to a perplexity degradation of only 0.08, which is smaller than that of Quamba2, despite quantizing more tensors—including $CB$, State, and $(X \cdot LUT)$—because axis-aware quantization more effectively minimizes error.

---

> ### Author Response · Authors · 2025-11-21
> **Response #2**
>
> **[W3] Citation or discussion on HadMamba**
>
> Thank you for pointing this out. HadMamba2 applies a Hadamard transform used in Quamba2 to the Mamba-2 projection weights, followed by GPTQ-based weight quantization, and then performs naive RTN quantization on the SSD inputs. In the revised version, we will properly explain this method, clearly describe how it differs from SSDi8, and also include the information highlighted in blue in the experimental setup.
>
> **[Q2] Latency breakdown of the entire Mamba-2 block**
>
> | Model (2.7B, L=2048) | Method | Batchsize | in_proj | conv  | ssd    | norm  | out_proj | correction |
> |----------------------|--------|-----------|---------|-------|--------|-------|----------|------------|
> |                      | FP16   | B=8       | 9.829   | 0.781 | 4.795  | 1.196 | 4.919    | —          |
> |                      | FP16   | B=16      | 20.759  | 1.357 | 9.426  | 1.640 | 10.272   | —          |
> |                      | FP16   | B=32      | 41.297  | 2.526 | 17.640 | 3.086 | 20.474   | —          |
> |                      | SSDi8  | B=8       | 8.124   | 0.587 | 4.481  | 1.089 | 3.266    | 0.426      |
> |                      | SSDi8  | B=16      | 16.758  | 1.011 | 7.293  | 1.442 | 6.677    | 0.698      |
> |                      | SSDi8  | B=32      | 33.893  | 1.799 | 13.199 | 2.291 | 12.914   | 1.252      |
>
> Thank you for the suggestion. We provide a detailed latency breakdown of the 55th block of the 2.7B Mamba-2 model under batch sizes $B=8,16,32$. As shown in the breakdown table, the dominant bottleneck in FP16 is the projection layer, followed by the SSD computation, while normalization and other components contribute minimally. In SSDi8, we apply Hadamard-based activation quantization and GPTQ weight quantization to the projection layers, and focus our primary optimizations on the SSD module, which constitutes the second-largest bottleneck. Across different batch sizes, the increase in SSD latency scales approximately linearly, similar to the projection layer, while other submodules remain negligible. We also note that the output projection reduces the dimensionality of the hidden representation, allowing the mean-correction term introduced by our SSDi8 to be applied at minimal cost. Its latency remains lower than that of even the convolution layer, which is the least expensive FP16 component. We will include this breakdown in the revised version to clarify the scalability of SSDi8.
>
> **[M1] Simplifying the shape conversion and tensor dimension**
>
> Following your suggestion, we simplified the explanation of shape conversions and tensor dimensions in Sec. 4.1 and added corresponding pseudocode in Appendix B.
>
> **[M2] Discussion on large-batch serving scenarios**
>
> | Batchsize |        FP16         |                  |        SSDi8        |                  |
> |-----------|----------------------|------------------|----------------------|------------------|
> |           | SSD Latency (ms)     | Throughput (T/s) | SSD Latency (ms)     | Throughput (T/s) |
> | 16        | 2.757                | 9918             | 2.671                | 10299            |
> | 32        | 4.898                | 10653            | 4.510                | 11976            |
> | 64        | 9.271                | 10405            | 7.173                | 12114            |
> | 128       | 18.003               | 11222            | 13.406               | 13407            |
> | 256       | 35.194               | 11333            | 23.730               | 13798            |
>
> Following your suggestion, we additionally evaluate SSDi8 under varying batch sizes, from edge-oriented small batches ($B=16$) to large-batch cloud-serving scenarios ($B=256$). For $B=16$, where computational intensity is relatively low, SSDi8 improves SSD latency by 15% and throughput by 3.8%. As batch size increases, the computational load of SSD grows, and the benefits of quantizing ChunkState and ChunkBMM become more pronounced. In the large-batch regime ($B=256$), SSDi8 achieves a 32.5% reduction in SSD latency and a 21.7% improvement in throughput. We will incorporate this discussion into the revised version to better reflect SSDi8’s behavior in both edge and cloud-serving deployment settings.

---

> ### Comment · Reviewer_9vyW · 2025-11-24
> **Reply to authors' rebuttal**
>
> Thanks for the clarification. However, I still find several experimental settings in the rebuttal unclear, and I have a few follow-up questions and comments:
>
> - I am confused by Table [W2, Q1] and its accompanying description. It appears that PIR is never actually applied in the table. Could you clarify whether PIR was used in these experiments?
> - The reported end-to-end throughputs also seem unrealistic. For example, achieving 12,943 tokens/sec for SSD-FP16 appears too high, especially considering that the Quamba2 paper reports a TPOT of 22.73 ms on an A5000 GPU with batch size 1. Also, given the batch size 32, Tables [W2, Q1] reports 12,943 tokens/sec but Table [M2] reports 10,653 tokens/sec. It would be good to explain the experimental settings.
> - The speedup of the linear layer reported in Table [Q2] seems marginal. What precision configuration did you use for this experiment (e.g., W8A8)?
>
> I think that SSD is an efficient algorithm for computing SSM, which is one of the operations in Mamba models. However, in most contexts, SSDi8 refers to the latency or accuracy of the entire model in the manuscript, if I understand it correctly. Also, except for the Int8 SSD, most of the operators are reused from Quamba2. Given this, would it make more sense to rename the SSDi8 rows to Quamba2+SSDi8 throughout the manuscript?

---

> ### Author Response · Authors · 2025-11-24
> **Official Comment by Authors**
>
> **[R1] Table [W2, Q1]**
>
> We apologize for the confusion caused by Table [W2, Q1]. In the original table, the ablation results for SSDi8 were mistakenly shifted by one column. Now we have corrected the table.
>
> To clarify, Quamba2 indeed provides an important first step in applying PTQ to Mamba-2. However, its design quantizes $X$ directly, and the resulting quantization--dequantization overhead makes it difficult for Quamba2 to incorporate PIR in practice. In contrast, SSDi8 applies the Sparse-Aware Reformulation (SAR), which avoids this overhead and therefore enables PIR to be used effectively. This distinction explains why SSDi8 achieves the larger latency improvements observed in the table.
>
> **[R2] End-to-end throughputs**
>
> We apologize that our experimental settings were not described with sufficient clarity. We provide a more detailed explanation below. First, the “SSD-FP16” configuration is consistent with the first row of Table 5 (ablation table) in our main manuscript. This setting uses the Mamba-2 2.7B model under the W4A8 quantization regime with sequence length $L=2048$ and batch size $B=32$, while keeping only the SSD layer in FP16. Consequently, its throughput is significantly higher than that of a fully FP16 model. For reference, under the same configuration (2.7B, W4A8, $B=32$, $L=2048$), the full FP16 model's throughput and latency are as follows:
>
> | FP16 PPL | FP16 SSD Latency (ms) | FP16 Throughput (T/s) |
> |----------|-------------------------|--------------------------|
> | 9.05     | 17.925                 | 11320                   |
>
> Second, the throughput in Table [W2, Q1] is reported under batch size $B=32$ with sequence length $L=2048$, whereas Table [M2] uses sequence length $L=512$ for the large-batch experiment. All experiments were conducted on an NVIDIA A5000 GPU. We again apologize for causing confusion.
>
> **[R3] Speedup of the linear layer**
>
> This experiment was conducted on the Mamba-2 2.7B model under the W4A8 configuration with batch sizes $B \in \{8,16,32\}$ and sequence length $L=2048$. The linear layers use 4-bit weight quantization via GPTQ and 8-bit activation quantization based on the Hadamard transform.
>
> **[R4] Renaming the SSDi8**
>
> Our SSDi8 framework applies the standard Hadamard+GPTQ--based quantization outside the SSD layer—an approach commonly used in prior work, including Transformers—and applies our SSD-aware quantization only inside the SSD layer, where our main contributions lie. In this sense, the components outside SSD are indeed influenced by prior methods such as Quamba1 and Quamba2.
>
> However, the central contribution of SSDi8 is fundamentally different from Quamba2. Whereas Quamba2 focuses on SSD quantization through a reordering-based strategy, SSDi8 reinterprets the internal computations within SSD and quantizes the full set of SSD modules, including the state transition, through sparse-aware reformulation, activation reuse, and a persistent INT8 execution path. This constitutes a substantially different design and quantization mechanism inside SSD, even though Quamba2 provides an important foundation for components outside the SSD.
>
> We fully acknowledge the significance of Quamba2 in this line of research, and we will clearly cite and acknowledge Quamba2 both in the revised manuscript and in the accompanying GitHub release.

---

> > ### Comment · Reviewer_9vyW · 2025-11-25
> > **Rely to authors**
> >
> > We thank the authors for the additional clarification. I recommend that the authors revise some terminology to avoid potential confusion, such as the use of notations like Q(SSD) and SSDi8. Reviewer akaA also raised similar concerns regarding the distinction between the proposed method and Quamba2. Therefore, I would also encourage the authors to update the Table 5 based on our discussions. In summary, I think that the proposed sparse-aware reformulation and the Int8 SSD with mean correction are interesting and somehow novel. Given the authors’ responses and the extra experiments provided, I will increase my score, while leaving the final decision to the ACs and PCs.

---

> > > ### Author Response · Authors · 2025-11-27
> > > **Official Comment by Authors**
> > >
> > > We sincerely thank you for your insightful and constructive feedback. We will refine the notation and terminology, and finalize the updates to Table 5 once all reviewers’ feedback has been consolidated. We truly appreciate your guidance and support.

---

### Official Review · Reviewer_mHsN · 2025-10-31

**Soundness:** 4
**Presentation:** 3
**Contribution:** 3
**Rating:** 8
**Confidence:** 4

**Summary:**

This paper presents a post-training quantization framework specifically designed for the Structured State Space Duality (SSD) layers in Mamba-2 models. They enable INT-8 quantization inside SSD blocks through a combination of: a sparse-aware reformulation that decouples element-wise multiplications from matrix multiplications, strategic placement of quantization operations to minimize overhead, and a mean correction mechanism. The authors achieve comparable accuracy to FP16 while delivering up to 1.4x speedup in inference. The work addresses the challenge that existing quantization methods designed for Transformers fail when applied to SSD layers due to their unique computational structure involving recurrent states and channel-varying activations.

**Strengths:**

- first work to successfully apply INT-8 quantization within Mamba-2's SSD architecture
- Thorough mathematical analysis
- comprehensive experimental validation across multiple model sizes up to 8B
- Deployment validation on resource-constrained devices (Orin Nano)
- clear identification and analysis of why existing methods fail on SSD layers
- well-structured paper with detailed ablation studies

**Weaknesses:**

- unclear notation and presentation around the reformulation (mixing step-by-step vs chunked vs parallel forward passes)
- limited theoretical justification for why the mean correction works
-some implementation details are vague (e.g., specific quantization schemes for different components)
- the paper could benefit from more intuitive explanations alongside the mathematical formulations
- limited discussion of potential limitations or failure cases
- not immediately clear what the overhead is of computing and storing the quantization scales for the various per-channel schemes

**Questions:**

1. Do you observe any mismatch between step-by-step vs. chunked vs. parallel forward passes, given that quantization introduces errors in the recurrence?
2. In Section 1, you claim to show results for Hadamard rotation and GPTQ but these are not in Table 1 or other results tables. Could you clarify where these comparisons are?
3. Why is the mean correction applied only to the output projection layer? What happens if applied elsewhere?
4. How sensitive is the method to the calibration dataset size and choice?
5. Could you provide more details on how the sparse-aware reformulation maintains numerical stability?
6. How does the method perform on longer sequences beyond $L=2048$?

---

> ### Author Response · Authors · 2025-11-21
> **Response #1**
>
> We appreciate the reviewer’s positive assessment of our work. As noted, this paper provides the first successful application of INT8 quantization within Mamba-2’s SSD architecture, supported by a thorough mathematical analysis and comprehensive experiments across model sizes up to 8B. We also thank the reviewer for highlighting our deployment results on resource-constrained hardware (Orin Nano), the clear diagnosis of why prior methods fail on SSD layers, and the clarity of our ablation design and presentation.
>
> We address all of your comments in the following responses:
>
> **[W1] Unclear notation and presentation around the reformulation (mixing step-by-step vs chunked vs parallel forward passes)**
>
> Based on your suggestion, we have expanded the explanation around the reformulation in Sec. 4.2, detailing the per-dimension computation flow and the justification for the reformulation. The corresponding revisions are highlighted in blue.
>
> **[W2] Limited theoretical justification for why the mean correction works**
>
> We appreciate the reviewer raising this point. Mean correction adjusts layer outputs by adding a correction term that compensates for quantization error. The correction term is derived as the optimal solution that minimizes the squared quantization error, and depending on whether it is scalar, vector, or matrix, it corresponds to the tensor mean, a row/column-wise mean vector, or a full residual tensor.
>
> In SSDi8, **we select the vector-form correction term to balance correction strength with memory and compute overhead.** Its derivation is provided as follows:
>
> Let $R = Y - Y' \in \mathbb{R}^{N \times P}$.
>
> The objective is
> $E_c = \lVert Y - (Y' + c) \rVert_F^2 = \lVert R - c \rVert_F^2 = \sum_{p=1}^P \sum_{i=1}^N (R_{i,p} - c_p)^2$.
>
> Since each column $p$ is independent:
> $E_p(c_p) = \sum_{i=1}^N (R_{i,p} - c_p)^2$.
>
> Differentiate w.r.t. $c_p$:
> $\frac{\partial E_p}{\partial c_p}
> = \sum_{i=1}^N 2(c_p - R_{i,p})
> = 2N c_p - 2 \sum_{i=1}^N R_{i,p} = 0$.
>
> Therefore,
> $c_p^\star = \frac{1}{N} \sum_{i=1}^N R_{i,p}
>           = \frac{1}{N} \sum_{i=1}^N (Y - Y')_{i,p}$.
>
>
> When applied, the operation follows $Y \rightarrow Y + c,$ and the resulting error satisfies $\||Y - (Y' + c)\||^2 \le \||Y - Y'\||^2,$ showing that **the corrected output never increases the quantization error:**
> Let $R = Y - Y' \in \mathbb{R}^{N \times P}$. For each column $p$, define the mean $\mu_p = \frac{1}{N}\sum_{i=1}^N R_{i,p}$. The objective for column $p$ is
> $E_p(c_p) = \sum_{i=1}^N (R_{i,p} - c_p)^2$.
> Completing the square gives
> $E_p(c_p) = \sum_{i=1}^N (R_{i,p} - \mu_p)^2 + N(\mu_p - c_p)^2$.
> Thus the minimum is achieved at $c_p^\star = \mu_p$, and
> $E_p(c_p^\star) = \sum_{i=1}^N (R_{i,p} - \mu_p)^2$.
> Without correction ($c_p = 0$), we have
> $E_p(0) = \sum_{i=1}^N (R_{i,p} - \mu_p)^2 + N\mu_p^2$,
> so
> $E_p(0) - E_p(c_p^\star) = N\mu_p^2 \ge 0$.
> Hence $E_p(c_p^\star) \le E_p(0)$ for every $p$.
> Summing over all columns, with $c^\star \in \mathbb{R}^P$ defined by $c^\star_p = \mu_p$, we obtain
> $\||R - c^\star\||_F^2 \le \||R\||_F^2$.
> Equivalently, since $R = Y - Y'$, the correction $Y' \leftarrow Y' + c^\star$ satisfies
> $\||Y - (Y' + c^\star)\||_F^2 \le \||Y - Y'\||_F^2$,
> so the post-correction error never increases.
>
> Since the correction uses only the layer output, it is applied consistently regardless of the internal computation of each module.
>
> **[W3] More intuitive explanations alongside the mathematical formulations**
>
> Thank you for the helpful comment. We have added a detailed mathematical explanation of the equations in Sec. 4.2 (highlighted in blue), as well as pseudo-code for Sec. 4.1 to facilitate understanding in Appendix B.
>
> **[W4] Limited discussion of potential limitations or failure cases**
>
> SSDi8 accounts for the characteristics of the internal parameters and operations within SSD, and by quantizing them appropriately to form an INT8 execution path, it achieves reductions in both latency and quantization error. For the in-projection and out-projection layers—which also contribute to end-to-end latency bottlenecks—we currently apply only the Hadamard+GPTQ method, which we did not further optimize in this work. We plan to explore more comprehensive quantization strategies for these components in future work.

---

> ### Author Response · Authors · 2025-11-21
> **Response #2**
>
> **[W5] Overhead of computing and storing the quantization scales for the various per-channel schemes**
>
> Thank you for the valuable suggestion. We quantize the following parameters inside the SSD layer, along with their corresponding quantization scales:
>
> $X$ (more precisely, $X \cdot LUT$): $[B, H, c, P, l]$ $\rightarrow$ quantized per $(H, P)$, Scale: $[H, P]$
>
> $B$ and $C$: $[B, G, c, l, N]$ $\rightarrow$ quantized per $G$, Scale: $[G]$
>
> $CB$: $[B, G, c, l, l]$ $\rightarrow$ quantized per $G$, Scale: $[G]$
>
> State: $[B, H, c, P, N]$ $\rightarrow$ quantized per $(H, P)$, Scale: $[H, P]$
>
> As shown in Fig. 2 of the main manuscript, $H \cdot P = D$, and in Mamba-2 8B, $G = 8$ and $D \approx 8\mathrm{k}$. Since all quantization scales are stored in FP16, the amount of quantization scales saved during calibration is about 0.03 MB per block. Therefore, from both the storage and computation perspectives, the overhead associated with the quantization scales is negligible.
>
> **[Q1] Any mismatch between step-by-step vs. chunked vs. parallel forward passes**
>
> | Chunksize | Mode          | FP16 PPL | FP16 SSD Latency (ms) | FP16 Throughput (T/s) | SSDi8 PPL | SSDi8 SSD Latency (ms) | SSDi8 Throughput (T/s) |
> |-----------|---------------|----------|------------------------|------------------------|-----------|--------------------------|--------------------------|
> | 1         | Parallel      |   OOM  | OOM                     |       OOM             |  OOM   | OOM                       |       OOM                  |
> | 32        |              | 9.05     | 16.332                 | 9415                   | 9.70      | 9.084                    | 11978                    |
> | 128       |              | 9.05     | 9.071                  | 11148                  | 9.50      | 7.281                    | 12985                    |
> | 256 (base)| Chunked       | 9.05     | 8.938                  | 11127                  | 9.48      | 6.800                    | 13067                    |
> | 512       |              | 9.05     | 10.762                 | 10754                  | 9.47      | 8.780                    | 12649                    |
> | 2048      | Step-by-Step  | 9.05     | 24.161                 | 8234                   | 9.43      | 15.457                   | 10349                    |
>
> Thank you for the clarification query. We interpret the reviewer’s “step-by-step,” “chunked,” and “parallel” forward passes as different choices of chunk size $l$, which determine the number of chunks $c$=seq_len/$l$. Accordingly, we vary $l$ from the parallel regime ($l=1$, no recurrence) to the step-by-step regime ($l=$seq_len, a single chunk) and report perplexity, SSD latency, and throughput.
>
> In FP16, perplexity remains identical across all settings, as expected, while latency increases as $l$ moves away from the default configuration due to tensor-shape changes and increased state interactions across chunks.
>
> Under SSDi8 quantization, we observe divergence across these regimes. Perplexity degrades as the number of chunks grows, with the parallel regime (small $l$, large $c$) showing the largest degradation. This aligns with our analysis: in the StatePassing module, the state tensor $(B,H,c,P,N)$ interacts across chunks, and repeated INT8 transformations accumulate quantization error when $c$ is large. Conversely, the step-by-step regime ($c=1$) yields the smallest error.
>
> For efficiency, SSDi8 reduces latency across all configurations and yields the largest improvements for large $l$. This occurs because larger chunks amplify the computational and memory cost of ChunkState and ChunkBMM, particularly for tensors such as $CB \in \mathbb{R}^{B,H,C,l,l'}$ that contain two $l$-dimensions. Quantizing these operations substantially mitigates the bottlenecks introduced by large-chunk computation.
>
> **[Q2] Results for Hadamard rotation and GPTQ**
>
> Thank you for pointing this out. In our paper, the baseline labeled HAD refers to the combination of all three components: (1) Hadamard rotation applied to the Mamba-2 projection weights, (2) GPTQ-based weight quantization, and (3) naive RTN quantization for the SSD inputs. Thus, the "Hadamard + GPTQ" comparison is not missing from Tab. 1; rather, it is represented by the HAD (or HadMamba2) method, which integrates all of these steps. We have added this definition to the experimental setup section, including the information highlighted in blue, to make it explicit and avoid confusion.

---

> ### Author Response · Authors · 2025-11-21
> **Response #3**
>
> **[Q3] Why is the mean correction applied only to the output projection layer?**
>
> Thank you for the clarifying question.
>
> | Model | Bitwidth | Quantized Layer(s) | ACC   |
> |-------|----------|--------------------|-------|
> | 2.7B  | FP16     | –                  | 63.8% |
> |       | W4A8     | + In Proj          | 63.6% |
> |       |          | + SSD              | 58.4% |
> |       |          | + Out Proj         | 54.6% |
>
> Because Mamba layers do not include biases, the correction term must be applied online, and its dimensionality is determined by the output dimension of the layer to which it is applied. The output projection layer has the smallest output dimension--approximately one quarter of the input projection--and, as shown in the table above, it also accumulates the largest quantization error. For this reason, we prioritized applying mean correction at the output projection.
>
> | Bitwidth | Project | Speedup | Acc.  |
> |----------|---------|---------|-------|
> | **W4A8** | None    | ×1.00   | 67.2% |
> |          | In      | ×0.945  | 67.4% |
> |          | SSD     | ×0.975  | 67.5% |
> |          | Out     | ×0.987  | 67.4% |
>
> The table above reports results when applying correction to the input projection, SSD, and output projection layers. The dimensional ratios of these layers are roughly \(4 : 2 : 1\), and the overhead of applying correction follows the same ratio. While the correction provides similar accuracy benefits across all three layers, the overhead differs substantially. Considering the trade-off between accuracy gain and computational cost, we therefore apply mean correction only at the output projection layer.
>
> **[Q4] Sensitive to the calibration dataset size and choice?**
>
> To assess sensitivity to the calibration dataset size, we vary the number of calibration samples from 128 to 2048 (a 16$\times$ increase) and report accuracy on six zero-shot tasks :
>
> | Calib Num      | Wino | Piqa | Arc C | Arc E | Hella | Lamb | Avg. |
> |-------------|------|------|--------|--------|--------|--------|--------|
> | 128         | 64.4% | 75.5% | 35.1% | 68.2% | 65.0% | 68.0% | 62.7% |
> | 256         | 64.5% | 75.3% | 36.4% | 67.8% | 64.8% | 67.0% | 62.6% |
> | 512         | 64.2% | 75.5% | 35.8% | 67.9% | 65.1% | 67.7% | 62.7% |
> | 1024        | 64.3% | 75.7% | 36.4% | 67.6% | 64.8% | 66.3% | 62.5% |
> | 2048        | 64.8% | 75.1% | 36.2% | 68.1% | 65.2% | 67.1% | 62.7% |
>
> Across this range, the average accuracy remains stable (62.5%-62.7%). These results indicate that SSDi8 is insensitive to the calibration dataset size.
>
> We also evaluate the effect of the calibration dataset by comparing three commonly used benchmarks: C4, PTD, and Pile. The results on six zero-shot tasks are shown below.
>
> | Calib Data | Wino | Piqa | Arc C | Arc E | Hella | Lamb | Avg. |
> |--------|------|------|--------|--------|--------|--------|--------|
> | C4     | 65.4% | 75.2% | 36.1%  | 68.5%  | 65.3%  | 67.7%  | 63%  |
> | PTD    | 65.1% | 75.1% | 36.5% | 68.6% | 65.5%  | 67.6%  | 63%   |
> | PILE  | 64.5% | 75.3% | 36.4%  | 67.8%  | 64.8%  | 67%    | 62.6% |
>
> Across the three datasets, the average accuracies are 63%, 63%, and 62.6%, respectively. While the variation across datasets is slightly larger than that observed when changing the calibration size, the fluctuation remains bounded within a comparable range.

---

> ### Author Response · Authors · 2025-11-21
> **Response #4**
>
> **[Q5] More details on how the Sparse-aware Reformulation maintains numerical stability**
>
> As shown in Appendix A, if the following inequality holds
> $\sum_{p=1}^{P} \frac{\Delta x_{p}^{2}}{12} \left( \frac{\Delta y_{p}}{\Delta x_{p}} \right)^{2} \cdot \Pr(y_{p} \neq 0) \le \lVert lut \rVert_{2}^{2} \sum_{p=1}^{P} \frac{\Delta x_{p}^{2}}{12}$,
>
> then $\mathrm{MSE}(x_{\mathrm{scaled}}) \le \mathrm{MSE}(x)$ holds.
>
> [Notation]
> - $x_p$: the $p$-th vector of the original input tensor $X$
> - $y_p$: the $p$-th vector of $y = x_{\mathrm{scaled}}$ obtained after applying the Sparse-aware Reformulation
> - $lut$: the scaling vector (lookup table) multiplied to each $x_p$ during reformulation
> - $\Delta_{x,p}$: quantization scale factor of $x_p$, $\Delta_{x,p} = \text{range}(x_p)/(2^{b}-1)$
> - $\Delta_{y,p}$: quantization scale factor of $y_p$, $\Delta_{y,p} = \text{range}(y_p)/(2^{b}-1)$
> - The delta ratio equals the ratio of ranges: $\Delta_{y,p}/\Delta_{x,p} = \text{range}(y_p)/\text{range}(x_p)$
>
> The satisfaction of this sufficient condition is determined by the following three factors:
> 1) the squared range ratio ($(\Delta_{y,p}/\Delta_{x,p})^2$)
> 2) the sparsity magnitude ($\Pr(y_p \neq 0)$)
> 3) the squared norm ($\lVert lut \rVert_2^2$)
>
> Our analysis of the actual SSD path shows that
> - the range ratio $\Delta_{y,p}/\Delta_{x,p}$ is generally less than 1,
> - the nonzero elements in $y_p$ are extremely sparse, so $\Pr(y_p \neq 0) \ll 1$,
> - the magnitude of $lut$ is typically greater than 1, satisfying $\max(|lut|) \le \lVert lut \rVert_2^2$
>
> Therefore, in the sufficient condition,
> $\sum_{p=1}^{P} \frac{\Delta x_{p}^{2}}{12} \left( \frac{\Delta y_{p}}{\Delta x_{p}} \right)^{2} \cdot \Pr(y_{p} \neq 0) \le \lVert lut \rVert_{2}^{2} \sum_{p=1}^{P} \frac{\Delta x_{p}^{2}}{12}$, the right-hand side tends to be much larger than the left-hand side. Thus, the combination of range reduction, sparsity, and the magnitude of the lookup table naturally satisfies the condition.
>
> In other words, when excluding the effect of outliers, the Sparse-aware Reformulation can be understood as operating in a way that secures numerical stability throughout the entire INT8 quantization process, and the experimental results on quantization performance also support our proof.

---

> ### Author Response · Authors · 2025-11-21
> **Response #5**
>
> **[Q6] Performance on longer sequences beyond $L = 2048$**
>
> | Length | Batch |         FP16         |                     |                 |        Quamba2        |                     |                 |          SSDi8           |                     |        |
> |--------|--------|----------------------|---------------------|-----------------|------------------------|---------------------|-----------------|-------------------------|---------------------|--------|
> |        |        | SSD Latency (ms)     | Throughput (T/s)    | PPL             | SSD Latency (ms)       | Throughput (T/s)    | PPL             | SSD Latency (ms)        | Throughput (T/s)    | PPL    |
> | 2k     | B=8    | 4.990                | 10480               | 9.062           | 5.313                  | 11850               | 9.549           | 4.507                   | 11981               | 9.470  |
> | 4k     |    B=8     | 9.299                | 10716               | 13.845          | 8.998                  | 12333               | 11.502          | 7.231                   | 12818               | 9.021  |
> | 6k     |  B=8       | 13.564               | 10922               | 113.407         | 13.363                 | 12648               | 100.438         | 10.367                  | 13038               | 8.943  |
> | 8k     |   B=8      | 17.867               | 11173               | 444.467         | 15.932                 | 13079               | 309.182         | 12.635                  | 13418               | 9.040  |
> | 10k    |  B=8       | 22.061               | 11167               | 1577.003        | 19.806                 | 13121               | 1253.093        | 15.827                  | 13804               | 9.122  |
> | 12k    |   B=8      | 27.058               | 10887               | 2650.420        | 23.531                 | 13079               | —               | 18.168                  | 13743               | 9.239  |
> | 14k    |   B=8      | 30.870               | 11362               | 3881.000        | 27.368                 | 13291               | —               | 21.138                  | 14200               | 9.360  |
>
>
> Thank you for raising this point. To address your concern regarding the modest end-to-end improvements over Quamba2 up to longer contexts, we implement an optimized convolution CUDA kernel to remove the remaining bottlenecks outside the SSD block. We report full end-to-end results using throughput (tokens/s), defined as the number of generated tokens divided by the end-to-end latency. We also include SSD-layer latency and perplexity for completeness.
>
> On an A5000 GPU, we evaluate the 2.7B Mamba-2 model with batch size 8 over sequence lengths ranging from 2k to 14k. Across all lengths, SSDi8 consistently outperforms both FP16 and Quamba2 in throughput and SSD latency, and the gap increases as the sequence length grows. At 14k tokens, SSDi8 improves throughput by +2400 tokens/s over FP16 (+19.5%) and reduces SSD latency by 12.7 ms (-42%). A notable observation is that FP16 Mamba-2 exhibits a sharp perplexity degradation beyond sequence length 4k, whereas SSDi8 maintains stable perplexity. Prior work [1] reports that long sequences induce numerical instability inside SSD; our results suggest that SSDi8 stabilizes these numerical effects, even outperforming Quamba2 in this regime. As further evidence, when keeping the SSD block in FP16 and quantizing only the other components, we observe substantial degradation at longer sequence lengths (e.g., PPL 14.924 at 4k, 132.518 at 6k, and 682.839 at 8k), performing worse than full FP16. Upon closer examination, we find that quantizing only the state tensor yields a PPL of 9.89 at a context length of 6k. This observation is in line with prior findings in [1], which identify the state tensor as a source of numerical instability at long sequence lengths, indicating that quantizing the state tensor may have helped attenuate this effect. If the reviewers are interested in a deeper understanding of this phenomenon, we would be glad to perform additional analysis upon request.
>
> [1] Mamba Modulation: On the Length Generalization of Mamba Models, NeurIPS 2025

---

### Official Review · Reviewer_akaA · 2025-11-01

**Soundness:** 2
**Presentation:** 2
**Contribution:** 1
**Rating:** 4
**Confidence:** 4

**Summary:**

This paper presents SSDi8, a post-training quantization method built specifically for Mamba-2’s Structured State Space Duality (SSD) blocks. Instead of applying generic Transformer quantization, SSDi8 restructures SSD computations so they can run almost entirely in INT8. It quantizes the group-wise activations B and C once and reuses them across modules, reformulates ChunkState to keep the matrix multiplications in low precision, and adds a small per-channel mean correction to offset quantization drift. The result is near-FP16 accuracy with up to 1.47× faster inference over FP16 and 1.38× over Quamba2, marking the first fully INT8 data path within the SSD architecture.

**Strengths:**

A key strength of the paper is its architecture-aware approach to quantization. Rather than applying generic Transformer-style methods, SSDi8 directly models the structure of Mamba-2’s SSD blocks and exploits their specific activation patterns.

The design choices are well motivated by empirical observations, such as head-wise activation variance and per-axis sparsity.

The experiments are comprehensive, covering multiple model scales, bit-width regimes, and hardware platforms, with consistent accuracy close to FP16 and up to 1.47× latency improvement

**Weaknesses:**

The paper overemphasizes the need for a fully continuous INT8 execution path through SSD without clearly explaining why this is critical for Mamba-2’s efficiency. While the idea of keeping all operations in INT8 sounds appealing, the authors do not show evidence that occasional FP16 operations meaningfully degrade throughput. The claim that the presence of an FP16 activation such as LUTstate “eliminates the efficiency of INT8 GEMM” is not well supported. A mixed-precision path would not inherently destroy INT8 performance; it would simply require limited casting or dequantization overhead. More quantitative profiling is needed to show where and how FP16 operations bottleneck the pipeline.

The novelty claim that SSDi8 is “the first post-training quantization framework specifically designed for SSD” is also overstated. Quamba2 already targets the SSD modules in Mamba-2 and applies structured activation quantization within them. The paper should clarify what makes SSDi8 distinct from existing SSD-aware methods, such as whether the main innovation is the sparse-aware reformulation, activation reuse, or the persistent INT8 recurrence path, rather than relying on the claim of being the first.

**Questions:**

1. Why is maintaining a fully continuous INT8 execution path so essential for SSD efficiency?

2. Do the authors have profiling data showing where FP16 operations become actual latency bottlenecks in SSD?

3. How much runtime is really lost when LUT_state state remains in FP16—can this be quantified?

4. Could a mixed-precision (INT8 + FP16) path achieve similar performance without full reformulation?

5. The paper claims that FP16 activations “eliminate” INT8 GEMM efficiency—what hardware-level evidence supports that?

6. In what specific ways does SSDi8 differ from Quamba2, which already quantizes SSD blocks in Mamba-2?

7. Is the main novelty the sparse-aware reformulation, the activation reuse strategy, or the persistent INT8 recurrence path?

---

> ### Author Response · Authors · 2025-11-21
> **Response #1**
>
> We appreciate the reviewer highlighting the strengths of our work. We are grateful for the recognition of our architecture-aware quantization design, the empirical motivations behind our choices—such as head-wise variance and per-axis sparsity—and the comprehensive experiments demonstrating FP16-level accuracy and up to 1.47× latency improvement across model scales, bit-widths, and hardware platforms.
>
> We address all of your comments in the following responses:
>
> **[W1, 2, Q1-4] Overemphasis on the need for a fully continuous INT8 execution path**
>
> | Ablation | SSD Latency (ms) | Throughput (T/s) | PPL |
> |----------|------------------|----------------------|------|
> | SSD (FP16)        | 8.32   | 10791.4 | 9.40  |
> | Ablation1               | 8.017  | 11328   | 9.43  |
> | Ablation2               | 7.952  | 10552.8 | 9.595 |
> | Ablation3               | 8.465  | 10812.2 | 9.44  |
> | Ablation3-1             | 7.961  | 10987.8 | 9.468 |
> | Q(SSD) - Q(BMM)  | 7.607  | 12877   | 9.475 |
> | Q(SSD) + Q(BMM)  | 6.815  | 13069   | 9.48  |
>
> | Method | Batchsize | in_proj | conv  | ssd    | norm  | out_proj | correction |
> |--------|-----------|---------|-------|--------|-------|----------|------------|
> | FP16   | B=8       | 9.829   | 0.781 | 4.795  | 1.196 | 4.919    | —          |
> | FP16   | B=16      | 20.759  | 1.357 | 9.426  | 1.640 | 10.272   | —          |
> | FP16   | B=32      | 41.297  | 2.526 | 17.640 | 3.086 | 20.474   | —          |
> | SSDi8  | B=8       | 8.124   | 0.587 | 4.481  | 1.089 | 3.266    | 0.426      |
> | SSDi8  | B=16      | 16.758  | 1.011 | 7.293  | 1.442 | 6.677    | 0.698      |
> | SSDi8  | B=32      | 33.893  | 1.799 | 13.199 | 2.291 | 12.914   | 1.252      |
>
> We appreciate the reviewer highlighting this concern. We agree that a continuous INT8 path is not universally required for all architectures; however, for Mamba-2, keeping SSD computations in INT8 is particularly important for end-to-end latency. As shown in the first table above, quantizing SSD to INT8 yields a 24% improvement in throughput over retaining SSD in FP16. The second table further shows that SSD is the second most time-consuming component in Mamba-2 (2.7B, L=2048), so its precision level has a direct impact on overall throughput.
>
> Within SSD, avoiding FP16 operations is crucial for reducing unnecessary dequantization overhead. Figure 4m in the main manuscript shows that our Persistent INT8 Representation (PIR) achieves a 1.77× speedup over FP16 and significantly reduces latency in the ChunkScan module—the dominant internal bottleneck. This improvement arises because PIR avoids FP16–INT8 transitions and, together with SAR (Sparse-Aware Reformulation), enables lower-cost INT8 computation without additional quantize/dequantize operations.
> To further clarify why occasional FP16 operations degrade throughput, we conducted additional ablations:
> Ablation1 (INT8 B, FP16 X → FP16 State): ChunkBMM is quantizable, but PIR cannot be used; latency plateaus at a much higher level.
> Ablation2 (INT8 B, INT8 X → INT8 State using Q(Q(B)·LUT)): PIR is enabled, but extra dequantization of Q(B) and re-quantization of Q(scaledB) introduce overhead, making it slower than Ablation1. This aligns with prior findings in QServe [2], which reports that dequantization operations are a major bottleneck in PTQ pipelines.
> Ablation3 (INT8 State via Q(B·LUT) without reuse): Although PIR is possible, multiplying B and LUT requires expanding the Group dimension into the Head dimension, increasing tensor size up to 80× and making quantization prohibitively expensive. Even with additional computation (Ablation3-1), the performance remains below that of SAR.
> These results consistently show that maintaining an INT8 path—without extra quantize/dequantize steps—is essential for maximizing SSD performance, and that SAR provides the configuration that achieves this with low overhead.
>
> **[W2, Q5] Hardware-level evidence support for eliminating INT8 GEMM efficiency**
>
> As noted in our response to [W1, 2, Q1–4] (above), QServe [2] identifies the dequantization step as a major source of overhead during quantization. Following this observation, we initially used the term “eliminate,” but we agree that it may convey an unnecessarily strong implication, so we have replaced it with “undermine.” The corresponding revision in the sparse-aware reformulation section (sec. 4.2) is highlighted in blue.
>
> [2] QServe: W4A8KV4 Quantization and System Co-design for Efficient LLM Serving, MLSys 2025

---

> ### Author Response · Authors · 2025-11-21
> **Response #2**
>
> **[W3] Overstatement on the novelty claim**
>
> We appreciate the reviewer bringing this up. While Quamba2 does quantize SSD modules, it does not maintain a fully INT8 execution path inside SSD, as several submodules and activation flows still fall back to FP16. In SSDi8, the state tensor remains in INT8 within SSD, enabled by our parse-aware reformulation, activation reuse, and a persistent INT8 recurrence path.
>
> We agree that describing SSDi8 as "the first PTQ framework for SSD" may be too strong, and we revised the phrasing in the main manuscript to instead highlight that our contribution lies in fully modeling SSD’s internal structure to enable a complete INT8 path, rather than in the claim of being the first - for example, by stating that "our method is the first to maintain a persistent INT8 path and fully model SSD’s internal structure during PTQ." The corresponding updates in Sec. 1 and the abstract of the main manuscript are highlighted in blue.
>
> **[Q6] Does Quamba2 already quantize the SSD blocks?**
>
> We apologize for any confusion that may have arisen. While Quamba2 does quantize the inputs to the SSD (which are subsequently dequantized back to FP16 inside the SSD), it does not quantize the internal modules of the SSD. While Quamba2 is the first to apply PTQ to Mamba-2, its approach differs substantially from SSDi8. Quamba2 performs clustering-based group quantization on the inputs, but dequantizes them immediately before SSD computation. Consequently, all GEMM operations inside SSD remain in FP16, and the repeated dequantization introduces non-trivial overhead (e.g., SSD latency becomes higher than FP16 at batch 16, L=2048 for the 2.7B model).
>
> In contrast, SSDi8 explicitly incorporates the internal structure of SSD and ensures that the state tensor and all intermediate activations remain in INT8 from creation to output, avoiding unnecessary quantize/dequantize transitions and enabling a persistent INT8 recurrence path. Furthermore, instead of relying on indirect clustering heuristics, SSDi8 applies direct axis-aware quantization by identifying statistically consistent dimensions within SSD (e.g., per-P/H for X and per-G for State), allowing both lower quantization error and higher accuracy. In summary, SSDi8 differs from Quamba2 not only in which tensors are quantized but also in how SSD internals are modeled and how a full INT8 execution path is maintained.
>
> **[Q7] Main novelty of SSDi8**
>
> In SSDi8, the novelty does not come from a single technique but from how Sparse-aware Reformulation (SAR) and activation reuse jointly enable a persistent INT8 recurrence path inside SSD—an execution pattern that has not been achieved by prior PTQ methods, including Quamba2.
>
> The activation reuse strategy ensures that tensors repeatedly used throughout SSD are quantized only once and remain in INT8 across multiple modules, reducing memory traffic and avoiding redundant quantization steps. SAR then prevents additional dequantization cost by allowing these reused parameters to be consumed directly in INT8 form. Importantly, SAR also avoids quantizing the tensor after its G→H expansion—an operation that can inflate tensor size by up to 80×—thus maximizing latency benefits. Beyond its practical effect, SAR is grounded in a formal result (Appendix A) showing that the proposed reformulation achieves lower quantization MSE than directly quantizing the element-wise product, providing a principled justification for its design.
>
> These components together maintain the INT8 state tensor across ChunkState, StatePassing, and ChunkScan, enabling a fully persistent INT8 execution flow inside SSD. While PIR includes auxiliary steps such as bit shifts, its effectiveness fundamentally relies on SAR and reuse. As shown in Fig. 4, this leads to 1.77× speedup along the PIR path and 1.47× latency reduction for the full SSD block—gains that cannot be obtained without these combined techniques.

---

### Official Review · Reviewer_Kz2V · 2025-11-01

**Soundness:** 3
**Presentation:** 3
**Contribution:** 4
**Rating:** 8
**Confidence:** 4

**Summary:**

The paper introduces SSDi8, a post-training quantization (PTQ) framework specifically designed for Mamba-2's Structured State Space Duality (SSD) architecture. The authors first identify key challenges that make naive PTQ methods unsuitable for SSDs, such as heterogeneous activation scales and the mix of element-wise and GEMM operations. To address this, SSDi8 introduces three core contributions: (1) a sparse-aware reformulation that pre-scales activations to maintain a persistent INT8 data path within the SSD block; (2) a "quantize-once, reuse" strategy for channel-varying activations \(B,C\) along the group axis to minimize DRAM traffic; and (3) a lightweight per-channel mean correction technique to mitigate quantization bias.

Empirically, SSDi8 demonstrates near-FP16 accuracy on Mamba-2 models up to 8B parameters. It achieves up to a 1.47x speedup for the SSD block compared to FP16 (1.38x over the Quamba2 baseline) and shows practical latency gains on edge hardware like the Orin Nano. The framework strategically keeps two submodules, ChunkCumsum and ChunkScan2, in FP16 to preserve stability. A theoretical proposition in the appendix further supports the sparse-aware reformulation, showing it can reduce quantization MSE under certain conditions.

**Strengths:**

- **Principled, SSD-Specific Design:** The work is well-motivated, starting from a clear analysis of why standard PTQ fails for Mamba-2. The proposed sparse-aware reformulation and persistent INT8 state are theoretically grounded (as shown in the Appendix) and effectively target the unique properties of the SSD architecture.
- **Consistent Accuracy Gains:** The method's effectiveness is demonstrated across a range of model scales (1.3B, 2.7B, 8B) with perplexity and accuracy metrics that are nearly on par with the FP16 baseline and competitive with or better than Quamba2. The inclusion of latency measurements on edge hardware (Orin Nano) highlights the practical deployability of the framework.
- **Low-Overhead Accuracy Recovery:** The per-channel mean correction method effectively improves accuracy with a negligible latency overhead (1-2%), offering a practical and efficient solution for mitigating quantization bias.

**Weaknesses:**

- **Marginal End-to-End Speedup Over Prior Work:** While the SSD block itself shows significant speedups (up to 1.47x), the reported end-to-end latency gains over the most relevant baseline, Quamba2, appear modest. This raises questions about the overall practical impact, as other parts of the model may become bottlenecks. A more detailed breakdown of end-to-end latency would help clarify the real-world benefits.
- **Limited Evaluation on Recent Mamba-2 Models:** The evaluation is confined to the base Mamba-2 models. The paper misses an opportunity to demonstrate the framework's relevance and scalability by applying it to recent, influential foundation models that incorporate Mamba-2, such as Nemotron-4-340B-Base, which use hybrid SSM-Attention architectures and operate on much longer contexts.

**Questions:**

1.  The evaluation is performed on context lengths up to 4k. I believe this is limited by the base Mamba-2 models. However, a key advantage of quantizing SSDs may become apparent when scaling the sequence length to a longer one. It might be good and interesting to see how the e2e latency gains are there at longer context (e.g., 8k, 16k or longer)

2. Can SSDi8 apply on mabma2-transformer hybrid models [1,2]?

[1] https://huggingface.co/nvidia/mamba2-hybrid-8b-3t-128k
[2] https://huggingface.co/nvidia/Nemotron-H-8B-Reasoning-128K

---

> ### Author Response · Authors · 2025-11-21
> **Response #1**
>
> We thank the reviewer for the careful assessment of our work and for highlighting the main strengths of SSDi8. We appreciate the reviewer’s recognition of our three key techniques—sparse-aware reformulation enabling a persistent INT8 path, a quantize-once-and-reuse strategy for group-wise activations, and lightweight per-channel mean correction—as well as the strong empirical and theoretical results, including near-FP16 accuracy on models up to 8B parameters, up to 1.47× speedup, and practical latency gains on edge hardware.
>
> We address all of your comments in the following responses:
>
> **[W1, Q1] Marginal end-to-end speedup over prior work (including longer context results)**
>
> | Length | Batch |         FP16         |                     |                 |        Quamba2        |                     |                 |          SSDi8           |                     |        |
> |--------|--------|----------------------|---------------------|-----------------|------------------------|---------------------|-----------------|-------------------------|---------------------|--------|
> |        |        | SSD Latency (ms)     | Throughput (T/s)    | PPL             | SSD Latency (ms)       | Throughput (T/s)    | PPL             | SSD Latency (ms)        | Throughput (T/s)    | PPL    |
> | 2k     | B=8    | 4.990                | 10480               | 9.062           | 5.313                  | 11850               | 9.549           | 4.507                   | 11981               | 9.470  |
> | 4k     |    B=8     | 9.299                | 10716               | 13.845          | 8.998                  | 12333               | 11.502          | 7.231                   | 12818               | 9.021  |
> | 6k     |  B=8       | 13.564               | 10922               | 113.407         | 13.363                 | 12648               | 100.438         | 10.367                  | 13038               | 8.943  |
> | 8k     |   B=8      | 17.867               | 11173               | 444.467         | 15.932                 | 13079               | 309.182         | 12.635                  | 13418               | 9.040  |
> | 10k    |  B=8       | 22.061               | 11167               | 1577.003        | 19.806                 | 13121               | 1253.093        | 15.827                  | 13804               | 9.122  |
> | 12k    |   B=8      | 27.058               | 10887               | 2650.420        | 23.531                 | 13079               | —               | 18.168                  | 13743               | 9.239  |
> | 14k    |   B=8      | 30.870               | 11362               | 3881.000        | 27.368                 | 13291               | —               | 21.138                  | 14200               | 9.360  |
>
> Thank you for raising this point. To address your concern regarding the modest end-to-end improvements over Quamba2 up to longer contexts, we implement an optimized convolution CUDA kernel to remove the remaining bottlenecks outside the SSD block. We report full end-to-end results using throughput (tokens/s), defined as the number of generated tokens divided by the end-to-end latency. We also include SSD-layer latency and perplexity for completeness.
>
> On an A5000 GPU, we evaluate the 2.7B Mamba-2 model with batch size 8 over sequence lengths ranging from 2k to 14k. Across all lengths, SSDi8 consistently outperforms both FP16 and Quamba2 in throughput and SSD latency, and the gap increases as the sequence length grows. At 14k tokens, SSDi8 improves throughput by +2400 tokens/s over FP16 (+19.5&#37;) and reduces SSD latency by 12.7 ms (-42&#37;). A notable observation is that FP16 Mamba-2 exhibits a sharp perplexity degradation beyond sequence length 4k, whereas SSDi8 maintains stable perplexity. Prior work [1] reports that long sequences induce numerical instability inside SSD; our results suggest that SSDi8 stabilizes these numerical effects, even outperforming Quamba2 in this regime. As further evidence, when keeping the SSD block in FP16 and quantizing only the other components, we observe substantial degradation at longer sequence lengths (e.g., PPL 14.924 at 4k, 132.518 at 6k, and 682.839 at 8k), performing worse than full FP16. Upon closer examination, we find that quantizing only the state tensor yields a PPL of 9.89 at a context length of 6k. This observation is in line with prior findings in [1], which identify the state tensor as a source of numerical instability at long sequence lengths, indicating that quantizing the state tensor may have helped attenuate this effect. If the reviewers are interested in a deeper understanding of this phenomenon, we would be glad to perform additional analysis upon request.
>
> [1] Mamba Modulation: On the Length Generalization of Mamba Models, NeurIPS 2025

---

> ### Author Response · Authors · 2025-11-21
> **Response #2**
>
> **[W2, Q2] Limited evaluation on recent Mamba-2 models (Mamba2-transformer hybrid models)**
>
> | Nemotron        | Wino | Piqa | Arc C | Arc E | Hella | Lamb | Avg. | PPL  | SSD latency | Forward latency |
> |-----------------|------|------|--------|--------|--------|--------|--------|-------|--------------|------------------|
> | FP16              | 73.8% | 80.9% | 55.8%  | 81.4%  | 80.6%  | 66.2%  | 73.1% | 8.42 | 19.834       | 109.873          |
> | INT8           | 73.5% | 80.7% | 55.9% | 81.5%  | 80.6%  | 66.3%  | 73.0% | 8.65 | 9.156        | 98.904           |
>
> Following your suggestion, we conducted experiments on Nemotron-H-8B-Reasoning by applying INT8 quantization to the SSD (selective scan) path, which is the portion compatible with SSDi8. Specifically, only the state-transition and associated linear operations inside the SSD module were quantized to INT8, while all remaining submodules were kept in FP16 to evaluate quantization performance.
>
> As shown in the table above, even with INT8 applied solely to the SSD region, zero-shot performance on Wino, Piqa, Arc C/E, Hella, and Lamb remains close to FP16. The average accuracy changes only minimally (FP16: 73.1% $\rightarrow$ INT8: 73%), and the PPL shift (8.42 $\rightarrow$ 8.65) is marginal, indicating that 8-bit quantization of the SSD path does not significantly impair the model’s generative quality. In contrast, the latency improvements from INT8 quantization of the SSD path are substantial: SSD-module latency decreases from 19.834ms to 9.156ms (approximately a 2$\times$ reduction), and the overall forward latency correspondingly decreases from 109.873ms to 98.904ms.
>
> Overall, applying INT8 quantization exclusively to the SSD path in Nemotron-H-8B-Reasoning significantly improves computational efficiency while keeping accuracy and PPL changes minimal. If desired, we can also conduct extending quantization to layers outside the SSD path.

---

> ### Comment · Reviewer_Kz2V · 2025-11-28
> **Response to Author's Rebuttal**
>
> Thank you to the authors for providing their responses.
>
> [Q1 Follow-up]. Can the author provide the reference to the PPL number mentioned? It seems not to be included in the attached table.
>
> [Q2 Follow-up]. I recommend that the author include these results in the appendix.

---

> > ### Author Response · Authors · 2025-11-30
> > **Official Comment by Authors**
> >
> > Thank you for the follow-up questions.
> >
> > [W1, Q1] The PPL values we mentioned come from additional long-context experiments that we ran after the original submission, so they do not appear in the earlier table.
> >
> > [W2, Q2] Thank you for the suggestion. We will include the Nemotron-H-8B-Reasoning results in the appendix in a future revision.

---

### Comment · Area_Chair_P4rn · 2025-11-24
**Please respond to authors' rebuttal**

Dear reviewers,

The authors have now posted their rebuttal. Please review it and submit your responses as soon as possible so that they still have adequate time to address any remaining questions or concerns. Please note that the discussion period between authors and reviewers will close on December 3, 11:59 PM AOE, after which no further comments can be exchanged.

@Reviewer 9vyW: Thank you for alreading having done this.

Best, Your AC

---

### Meta-Review · Area_Chair_n53Z · 2025-12-27

**Summary:**

1. Need more careful analysis of latency and bottlenecks to justify the claim that a fully continuous INT8 execution path in the SSD block is important for getting the best performance. This analysis should also clarify the differences between Quamba2 and SSDi8, as well as with HadMamba. [Kz2V, akaA, mHsN, 9vyW]

2. The paper should include results from more recent models that employ Mamba-2 in a hybrid configuration, such as Nemotron-4-340B-Base. [Kz2V]

3. It would be good to compare SSDi8 to other algorithms at longer context lengths. [Kz2V, mHsN]

4. Is the main novelty the sparse-aware reformulation, the activation reuse strategy, or the persistent INT8 recurrence path? [akaA]

5. The presentation of the sparse-aware reformulation needs to be clearer. Perhaps derive Equation (7) explicitly from the original ChunkState formulation to establish a clearer connection between the reformulated SSD equations and the preceding context. More details on how the sparse-aware reformulation maintains numerical stability would help. [9vyW, mHsN]

6. Explain the mean correction more clearly: explain implementation details, provide theory showing why it works, explain why is is only applied to the output projection. [mHsN]

7. How sensitive is the method to the calibration dataset size and choice? [mHsN]

8. Discuss potential limitations or failure cases. [mHsN]

**Reviewer Concerns:**

1. This set of related concerns was addressed in the rebuttal and revision. The revised paper includes an ablation study illustrating the impact of the fully continuous INT8 execution path (Table 5); a discussion of the differences between SSDi8 and Quamba2; and latency, perplexity, and downstream task performance comparisons between SSDi8, Quamba2, and HadMamba.

2. This concern was addressed in the rebuttal, but is not yet addressed in a revised version of the paper. The authors provided results for Nemotron-H-8B-Reasoning in the discussion with the reviewers and promised that the results would be included in a future revision of the paper, but those results are not yet present.

3. This concern was addressed in the rebuttal, but is not yet addressed in a revised version of the paper. The authors provided longer context results in the discussion with the reviewers, but those are not yet present in a revision of the paper.

4. This question was answered in the discussion: "In SSDi8, the novelty does not come from a single technique but from how Sparse-aware Reformulation (SAR) and activation reuse jointly enable a persistent INT8 recurrence path inside SSD—an execution pattern that has not been achieved by prior PTQ methods, including Quamba2."

5. This concern was addressed in the rebuttal and revision. The primary changes are to Section 4.2 and Appendix A.

6. This concern was addressed in the rebuttal and revision, primarily in Appendix B.

7. This concern was addressed in the rebuttal, but is not yet addressed in a revised version of the paper. The authors provided in the discussion an illustration of the effects of changing the size of the calibration set, showing that the performance of the method is not particularly sensitive to the calibration set size.

8. This concern was addressed in the rebuttal, but not really addressed by the revision.

**Reviewer Scores:**

Kz2V - This reviewer was already recommending accept as a poster, and I do not believe that they would have changed their recommendtation based on the discussion.
akaA - The authors addressed the most significant concerns of this reviewer, and I predict they would have increased their score.
mHsN - This reviewer was already recommending accept as a poster, and I do not believe that they would have changed their recommendtation based on the discussion.
9vyW - This reviewer stated that they were increasing their score somewhat.

---

### Decision · Program_Chairs · 2026-01-26

Accept (Poster)